# A new early actinopterygian from the Mid-Pennsylvanian Logan Quarry Shale member of Indiana

Chenchen Shen 1,2*

1 Department of Ecology and Evolutionary Biology, University of Kansas, Lawrence, Kansas, United States of America, 2 Biodiversity Institute, University of Kansas, Dyche Hall,Lawrence, Kansas, United States of America

* c721s275@ku.edu

## Abstract

*Kalops* is a genus of early ray-finned fish in North America. It is distinguished from early actinopterygians by the combination of the following characters: subterminal mouth, rounded oval shaped lacrimal, multiple small supraorbitals and suborbitals. So far, it has been only excavated from the Serpukhovian (Late Mississippian) Bear Gulch Limestone in Montana. Here, a new species of this genus, *Kalops loganensis* n. sp., is described based on two specimens from the Moscovian Logan Quarry Shales in Indiana. The phylogenetic analysis indicates that the new species is a member of *Kalops* and recovered as the sister of the other two species, and the genus is recovered as a basal Actinopterygii, as the sister group of some other genera from the Bear Gulch Limestone. This new species is the first record of *Kalops* outside of Bear Gulch. It provides evidence of survivability of this genus, and as the youngest member, proves that *Kalops* lived at least through the Mid-Pennsylvanian in North America.

## Introduction

### Taxonomic history of *Kalops*

The Carboniferous is the Late Paleozoic period when the marine ecosystem recovered from the mass extinction events at the end of the Devonian, which led to the extinction of about 90% of fish families [1]. Recent studies suggest that the marine vertebrate ecosystem had turned over and established a high diversity of both bony and cartilaginous fishes by the end of the Mississippian (Early Carboniferous) [1–3]. Studies on Serpukhovian fossil localities across North America and Europe suggest high richness in marine vertebrates [4–6]. Among them, one fossil Lagerstätte in North America, the Bear Gulch in Montana, has preserved an unmatched high diversity of vertebrate fossils of the Serpukhovian (Late Mississippian). To date, there are 141 studied species of vertebrates (55 Osteichthyes and 84 Chondrichthyes)

**Data availability statement:** All relevant data are within the manuscript and its Supporting Information files.

**Funding:** This work is supported by 2020, 2021, and 2022 Research Grants of the Association of Earth Science Clubs of Greater Kansas City for graduate students; 2021 BI Panorama Grant of the Department of Ecology & Evolutionary Biology, University of Kansas (student competition); 2022 and 2023 EEB Summer Funding for graduate students, University of Kansas. The funders has no role in study design, data collection and analysis, decision to publish, or preparation of the manuscript. The funding was for support my student PhD dissertation.

**Competing interests:** The authors have declared that no competing interests exist.

recovered from the Bear Gulch [7]. Among the vertebrates, the chondrichthyan species identified from Bear Gulch have also been identified from additional coeval localities [8], such as the Euro-North American *Harpagofututor volsellorhinus* [9], and *Thrinacoselache gracia* [10]. On the other hand, many actinopterygians excavated from the Bear Gulch, are restricted to this locality only, including the families Aesopichthyidae [11], and Guildayichthyidae [12], as well as the genus *Kalops* [13].

So far, there are two species of *Kalops* described, *K. monophrys* and *K. diophrys*, and both are recovered from the Bear Gulch Limestone [13]. This genus is characterized by an elongated body form, subterminal mouth with the oral margin of the premaxilla facing posteroventrad, dorsal and ventral median rostral bones, short dermosphenotic, multiple small supraorbitals, infraorbitals, and suborbitals. There are three most distinguishable features of *Kalops* than other "palaeoniscoid" fish: numerous supraorbitals, numerous suborbitals, and an accessory row of extrascapulae anterior to the main row. However, compare to the anatomical information on *Kalops*, we know very little about the geographical and time distribution of this genus [13].

## Geology and paleoenvironment of Black Shales

The mid-Pennsylvanian Black Shales in North America are widespread and combined with multiple regional geographical units across today's today's Oklahoma, Kansas, Missouri, Nebraska, Iowa, Illinois, and Indiana [14]. The 1–2-meter-thick shale is the evidence of the North American Mid-continent Sea (NAMS), which existed from Mid-Pennsylvanian to Early Permian, with the largest extension about 2,100,000 km$^2$, nearly twice as large as today's Hudson Bay [15]. The Black Shales in western Indiana, referred as the shallow sea of the Illinois basin (Fig 1), are concentrated in three geological units. Two of the units, the Mecca Quarry Shale member of the Linton Formation overlayed on coal IIIA (Colchester Coal) and the Logan Quarry Shale member of the Staunton Formation below coal III (Seelyville Coal), are from the Parke County, Indiana (Fig 2). Both are dated as Moscovian in the Mid-Pennsylvanian (early Westphalian in the northwest European system) (Figs 3) [16–19]. The Logan Quarry Shale member contains relatively complete and more delicately preserved actinopterygian specimens than the Mecca Quarry Shale member.

The Logan Quarry is one of the major localities of the Black Shales in Parke County, Indiana. The marine vertebrate fossil-bearing section is also within the Logan Quarry Shale member [16]. For the 46–55 cm thick Logan Quarry Shale member here, the vertebrate fossils are preserved in the 3.4 to 3.5 cm thick black shale (Level J) of the Logan Quarry Shale member [16].

The paleoenvironment of the Logan Quarry Shale member was firstly reconstructed as a combination of near shore marine and fresh water because the fossil content includes terrestrial plants (especially leaves), invertebrates (annelids, mollusks and brachiopods) and vertebrates [16]. This reconstruction was accepted by some later studies [24,25]. The primary causation of the fragmented and disarticulated fossils in the Black Shales was hypothesized as predation [16].

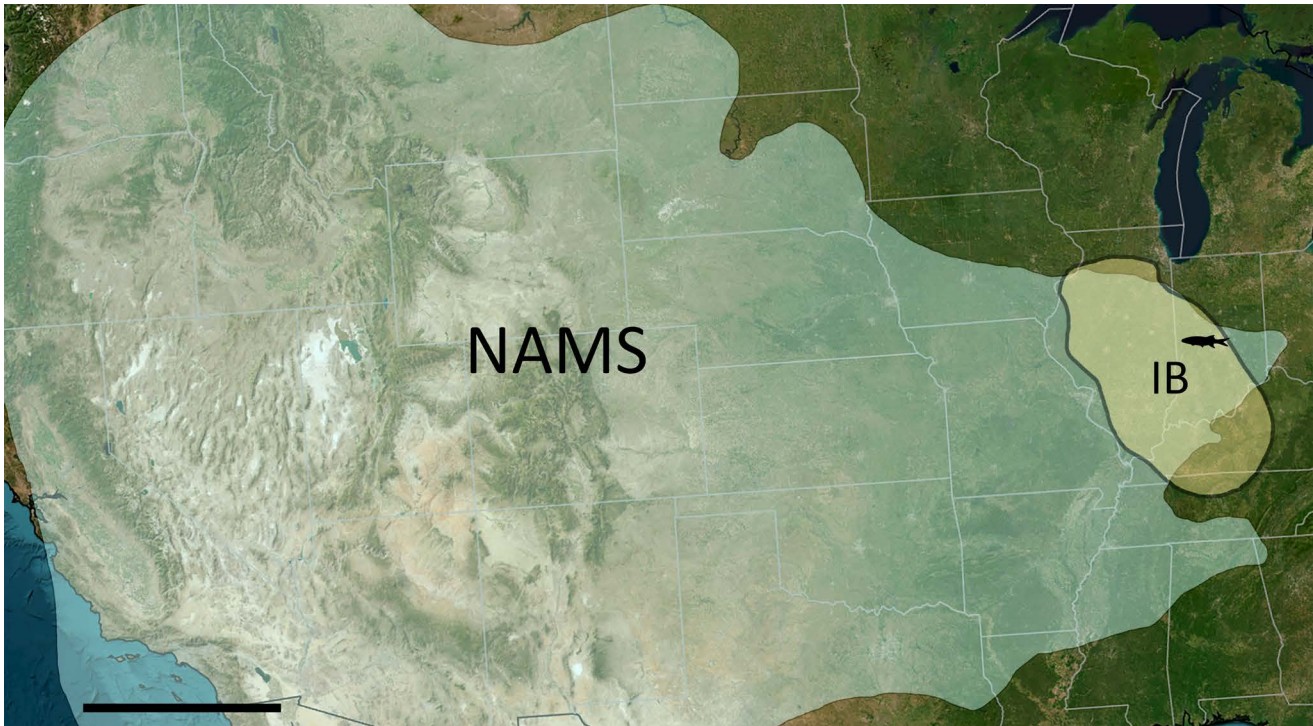

**Fig 1. Map of the North American Midcontinent Sea, approximately 308-309 Ma, in relate to today's Midwest area.** The light-blue highlighted area indicates the North American Midcontinent Sea. The light-yellow highlighted area indicates the Illinois Basin. The little fish marks the Logan Quarry. The black diagonal line marks the paleoequator. Abbreviations: **IB**: Illinois Basin; **NAMS**: North American Midcontinent Sea. Figure modified from https://eol.jsc.nasa.gov/ExplorePhotos/?illum=day. Scale bar = 500 km.

However, both hypotheses are questioned by more recent research. Studies of the fossil fauna and flora of the Upper Pennsylvanian Cohn Coal member in Illinois [26] and Joggins Formation in Nova Scotia in Canada [27,28], which have similar sedimentary facies as the Logan Quarry Shale member, show that such fossil combination of the sediment is an evidence of brackish water environment [29]. The original hypothesis about the fossil fragmentation in the Black Shales was questioned in a taphonomy study in 1984 [30], which also proposed that scavenging of the floating dead bodies is the dominant mechanism. The Logan Quarry Shale member is rich in vertebrate fossils including actinopterygians which are documented as "Palaeoniscoidea"; chondrichthyans (e.g., *Petrodus*, *Listracanthus*), and pleuracanthid; acanthodians fragments; and sarcopterygians (possible rhipidistian) [16,17]. Among them, cartilaginous fish fossils such as *Deanea meccaensis* [31], *Rhombacanthus zangerli* [32], *Ornithoprion hertwigi* [32], and *Cobelodus aculeatus*, have already been recognized and studied [33]. With the exception of Poplin [34], who described a specimen with a long rostrum based on X-ray radiograph, actinopterygians from the Logan Quarry have not been closely studied. The study of fossil actinopterygians from the Logan Quarry is critical to understanding the biodiversity of the Black Shales. Accordingly, fossil actinopterygian fish are studied and presented in this contribution.

A new species of *Kalops* based on two laterally preserved specimens collected from the mid-Pennsylvanian (Late Carboniferous) Black Shales unit, Staunton Formation in Logan Quarry, Indiana, is described. The new material extend the known time range of *Kalops* to Mid-Pennsylvanian, and slightly expands its geographical distribution to the eastern boundary of the NAMS.

| SGCS | | | Western Europe | USA |
|---|---|---|---|---|
| sub system | series | stage | stage | stage |
| Pennsylvanian | Upper | Gzhelian | Stephanian | Virgilian |
| | | Kasimovian | | Missourian |
| | Middle | Moscovian | Westphalian | Desmoinesian |
| | | | | Atokan |
| | Lower | Bashkirian | Namurian | Morrowan |

Fig 2. **The Pennsylvanian (Late Carboniferous) chronostratigraphic chart, modified from Lucas et al.** [20]. The arrow marks the age of the Logan Quarry.

## Materials and methods

**Material.** The studied fossil *Kalops* specimens from the Logan Quarry are housed in the Field Museum of Natural History (FMNH), Chicago, Illinois with the following catalogue numbers: FMNH PF- 2286, and FMNH PF-5919.

**Methods.** All specimens were already well prepared before this study started. No further preparation was conducted. The dusting of the fossils with ammonium chloride was made to facilitate observation of structures and photographing of specimens that are naturally black. Line drawings of each specimen were executed under a ZEISS 47 50 52 microscope with a camera lucida attachment, then final figures were prepared with Adobe Photoshop. All the photographs were taken by the author. The photographs of full body, skulls, and fins were taken with a Sony A-73 camera under natural light or electric

| state / stage | IN | KS | IL | NM |
|---|---|---|---|---|
| Desmoinesian | | | St. David m. | Gray Mesa Fm. |
| | Houchin Creek m. | Marmaton Gr. | Hanover Limestone m. | |
| | Excello Shale m. | | | |
| | Coal IV (Survant) | Bevier Fm. (Cherokee Gr.) | Purington Shale m. | |
| | Mecca Quarry Shale m. | Verdigris Fm. | Francis Creek Shale m. | |
| | Coal IIIa (Colchester) | | Colchester m. | |
| | Coal III (Seeleville) | Robinson Branch Fm. | | |
| | Logan Quarry Shale m. | Tebo Fm. | Vergennes Shale m. | |

**Fig 3. Approximated stratigraphic correlation of Desmoinesian stage among Indiana, Kansas, Illinois, and New Mexico states.** The information on regional stratigraphy units and their correlations are based on multiple previous works including [18,19,21–23]. The horizon of the Logan Quarry Shale member is highlighted in yellow. Abbreviations: **Fm**: formation; **Gr**: group; **m**: member.

light sources. The photographs of the individual scales were taken with an Infinity 2 camera, through a Leica M205C microscope, and captured via the Mac based Infinity Capture (version 5.1.2.40).

The method to generate the scale formula follows Westoll [35], which is to count of the number of scale rows in front of the origins of the fins, present as $\frac{D}{P\ A\ C}T$. In the scale formula, D is the number of scale rows before the origin of the dorsal fin, P is the number of scale rows before the origin of the pelvic fin, A is the number of scale rows before the origin of the anal fin, C is the number of scale rows before the origin of the caudal fin, and T is the number of scale row before the hinge line in the caudal peduncle.

**Nomenclatural Acts.** The electronic edition of this article conforms to the requirements of the amended International Code of Zoological Nomenclature, and hence the new names contained herein are available under that Code from the electronic edition of this article. This published work and the nomenclatural acts it contains have been registered in ZooBank, the online registration system for the ICZN. The ZooBank LSIDs (Life Science Identifiers) can be resolved and the associated information viewed through any standard web browser by appending the LSID to the prefix ""http://zoo-bank.org/"". The LSID for this publication is: urn:lsid:zoobank.org:pub:6A3621F4-A308-4B42-B6FA-8439C4505F62. The electronic edition of this work was published in a journal with an ISSN, and has been archived and is available from the following digital repositories: PubMed Central, LOCKSS.

**Phylogenetic method.** Giles et al. [36] performed a comprehensive phylogenetic analysis of early Osteichthyes by using 265 morphological characters, and both *K. monophrys* and *K. diophrys* were included in the analysis. The phylogenetic analysis performed here was conducted by adding the new species into the matrix of Giles et al. [36]. The settings of the characters from Giles et al. [36] are kept the same in this analysis, so that all characters are equally weighted, 258 characters are unordered, whereas seven characters are ordered (characters 89, 91, 151, 171, 241, 246, and 251). For the

list of characters, see Supplementary Material 1 (S1 Fig). *Acanthodes bronni* is selected as the outgroup. The parsimony analysis was conducted using the windows-based version of PAUP (4.0a, build 169).

**Terminology.** The terminology used for skull bones based on homology follows Schultze [37]. The traditional actinopterygian skull terminology is shown in parentheses after the corrected ones, such as postparietal (= parietal). The terminology for complex bones in the snout region follows Mickle [38,39], and for the dermosphenotic follows Poplin [40]. The terminology for scales follows Schultze [41–43] and the squamation regions follow Esin [44] (Fig 4). The terminology for fin rays, scutes and fulcra follows Arratia [45,46].

## Results

**Systematic Paleontology**

OSTEICHTHYES Huxley [47]

ACTINOPTERYGII Cope [48]

*Kalops* Poplin & Lund [13]

**Type Species.** *Kalops monophrys* Poplin & Lund [13].

**New Species:** *Kalops loganensis* n. sp.

   urn:lsid:zoobank.org:act:A8A6EAE2-DBD1-4040-96AD-8F24F1267EE9.

**Etymology.** After Logan Quarry, Parke County, Indiana, U.S.A. where the specimens were excavated.

**Diagnosis (based on a unique combination of characters).** A kalopsid fish of about 160mm long; with a fusiform body and a sharp snout; rostral region includes median dorsal rostral, median ventral rostral, and paired premaxillae; elongated parietal; postparietal is shorter than parietal; four extrascapulae, two on each side of the skull; the lacrimal is a distinct and separate bone, elongated and oval shaped; three small infraorbitals between jugal and dermosphenotic; the rectangular shaped dermosphenotic is short and deep; no dorsal dermosphenotic (= intertemporal); five small suborbitals in a vertical row between preopercle and infraorbitals; eight small supraorbitals in a horizontal row, forming the dorsal margin of the orbit; the oral margin of the upper jaw is formed by premaxilla, lacrimal and maxilla; maxilla shorter than the lower jaw; maxilla has a deep posterior plate and a deep posteroventral process; the anterior margin of the maxilla is ventral to

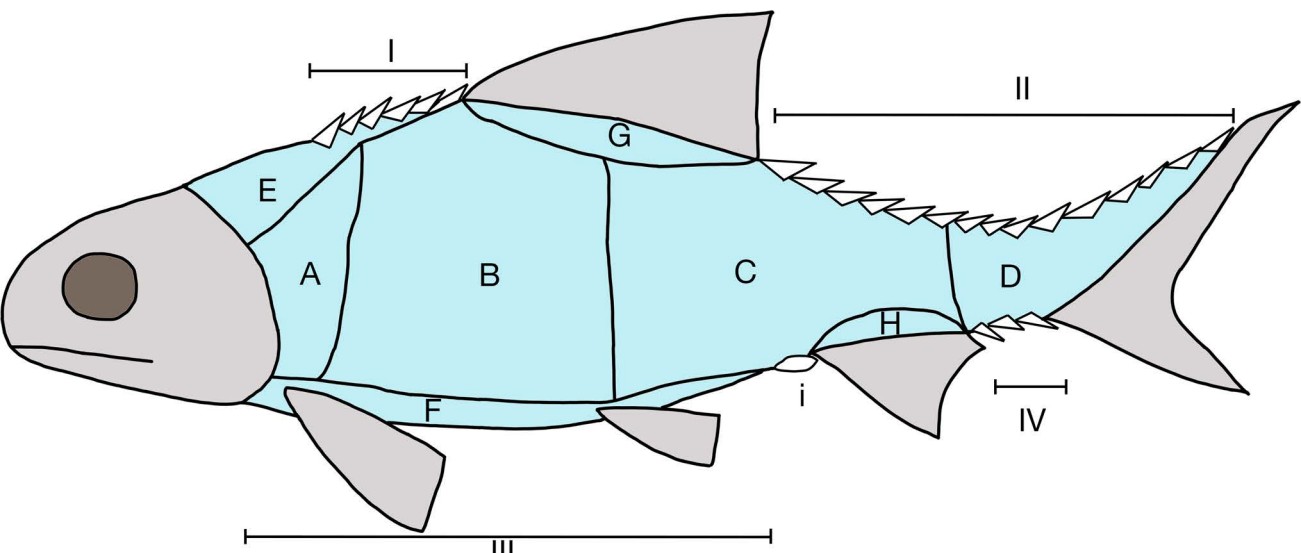

**Fig 4. Actinopterygian scale regions as suggested by Esin [44, fig. 2].** A-G are the lateral flank scales; I-IV are the ridge scutes; "i" is preanal scute.

the orbit; nine branchiostegal rays; body fully covered with rhombic scales with ganoine; the dorsal and ventral ridges of the body are covered by scutes; the posterior edge of the scales are straight; the predorsal ridge scutes have the same size as the scale rows ventral to them; all scales and scutes are heavily ornamented by continuous ridges of ganoine, the anterior ends of these ridges are finely crenulated.

Several features distinguish the new species from the Bear Gulch *Kalops,* including the absence of median extrascapular and of presupracleithrum. Additionally, all infraorbitals are of similar size.

**Holotype.** FMNH PF-2286, an almost complete specimen preserved as part and counterpart. The pelvic and anal fins are not preserved, and the caudal fin is only partially preserved in the hypaxial lobe.

**Referred specimen:** FMNH PF-5919, a nearly complete specimen. The anterior part of the skull, the pelvic fin, the anal fin, and the caudal fin are not preserved.

**Type locality.** Mid-Pennsylvanian Staunton Formation, Logan Quarry Shale member, collected from Level J in the Logan Quarry, Parke County, Indiana.

## Anatomical description

**General morphology and size**. This taxon is a fusiform fish. Both the FMNH PF-2286 and the FMNH PF-5919 specimens are incomplete, lengths of each specimen based on what is preserved were taken and used to estimate standard lengths. FMNH PF-2286 is 160mm long from the snout to the posterior, before the epaxial base of caudal fin, whereas FMNH PF-5919 is 150mm long, including the posterior half of the skull and the body trunk until the squamation hinge line. Though FMNH PF-2286 seems longer, FMNH PF-5919 is more incomplete and missing portions anteriorly and posteriorly. Based on this, it is estimated that the standard length of FMNH PF-5919 would have been slightly longer than FMNH PF-2286. In FMNH PF-2286 the head length is 45mm, relatively short compared to the estimated length of this specimen. The fish has a sharp snout, short preorbital length and orbital diameter, about 11% and 22% to the head length. The dermal bones of the skull are covered with ganoine. The body is fully covered by ganoid-type of scales. The caudal fin is heterocercal. (Fig 5). The scale formula is:

$$\frac{D\,20\,or\,23}{P\,10\,A?\,C\,37\,or\,42}\,T?.$$

**Snout**. The snout of *Kalops loganensis* is the same as *K. monophrys* and *K. diophrys*. The snout is sharp, and it is formed by a median dorsal rostral, a median ventral rostral, paired premaxillae, and nasal bones (Fig 6).

The median dorsal rostral is incomplete and only preserves its ventral part. This bone sutures with the nasal bones laterally and separates the nasal bones from contacting each other, and the median ventral rostral ventrally. The notch for the anterior nostril is on the lateroventral border of the median dorsal rostral. The median ventral rostral is shallow and wide, about twice the width of the median dorsal rostral. It is almost rectangular shaped. Its dorsal margin contacts both the dorsal rostral and the nasal bone, the posterior margin contacts the lacrimal (= infraorbital one), and the ventral margin contacts the premaxillae. The ventral margin is slightly convex. The ethmoid commissure is placed horizontally across the midline of the median ventral rostral. The nasal bone is deep, narrow and triangular in shape. It is deeper than the dorsal rostral. It contacts the parietal anterodorsally, and the first supraorbital posteriorly. The nasal bone does not connect with the dermosphenotic. The posterior nostril is located at the posteroventral border of the nasal bone. It is slightly more dorsal than the anterior nostril. All these bones are heavily ornamented by continuous and connected ridges of ganoine. The ornamentation ridges in dorsal portion of the nasal bone are longer and more vertically placed, whereas the ridges are shorter and horizontally placed in the ventral portion. In the nasal bone the supraorbital sensory canal is placed almost in a straight line.

The premaxilla is small, shallow and elongated antero-posteriorly. The posterior margin of this bone contacts the lacrimal posteriorly. There are no teeth preserved on the ventral margin of the premaxilla.

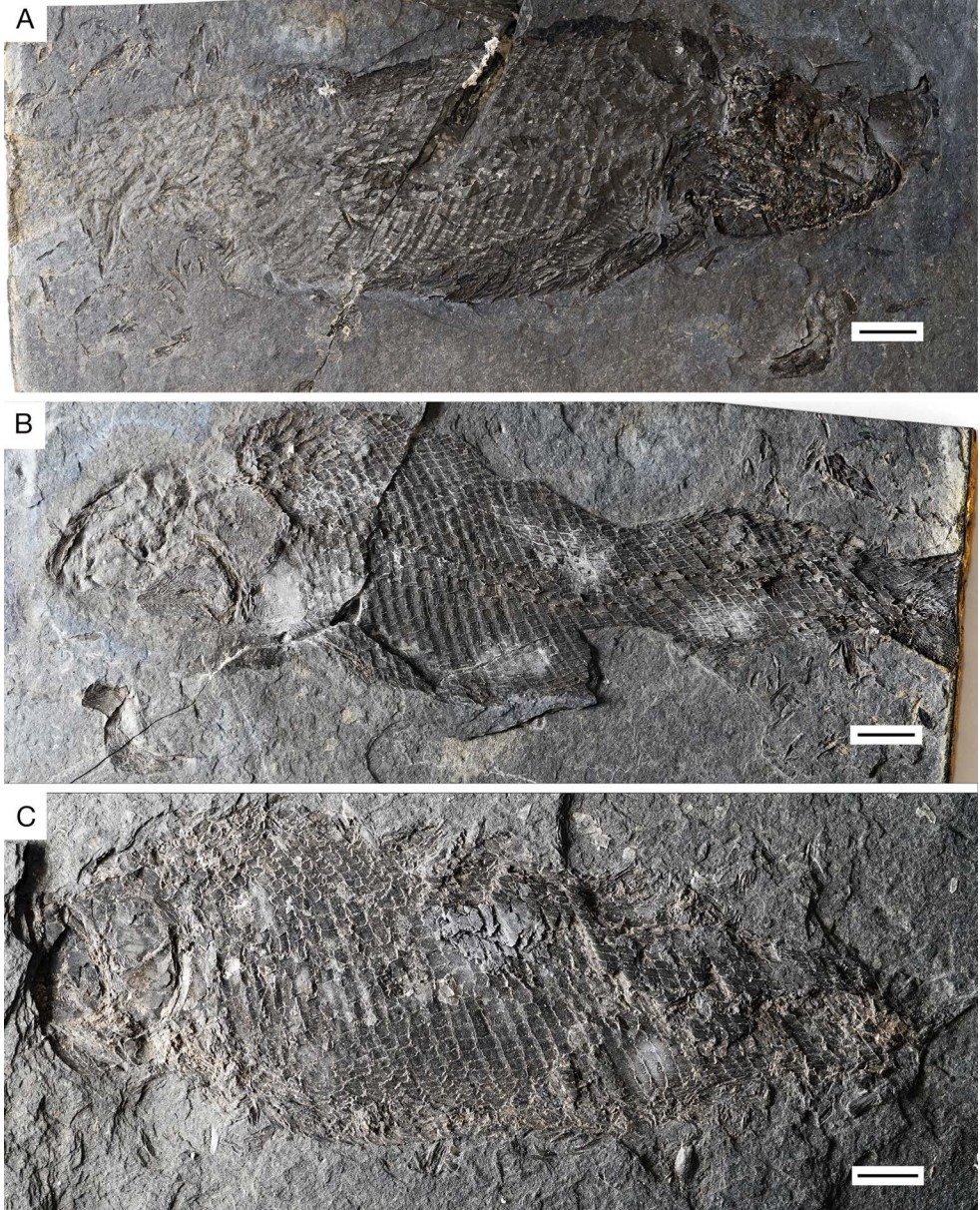

**Fig 5. *Kalops loganensis* n. sp.** **A.** Photograph of the holotype FMNH PF-2286, part; **B.** Photograph of the holotype FMNH PF-2286, counterpart; **C.** Photograph of specimen FMNH PF-5919. Scale bar = 10mm.

**Skull roof.** The skull roof is elongated and drastically expands laterad at the position of the supratemporotabulars (= dermopterotics). The region includes the parietal (= frontal), the postparietal (= parietal), the supratemporotabular, and the extrascapular bones (Fig 6).

The parietal is lost in FMNH PF-5919 and is only partially preserved in FMNH PF-2286. It is an elongated bone, about 34% of the skull length and its own length/width ratio is 2.8:1. This bone is sutured with the dorsal median rostral anteriorly, the nasal, the dermosphenotic, the supratemporotabular, and potentially the supraorbitals one to five laterally, and the postparietal posteriorly. Due to the preservation, the type of sutures, as well as the position of the pineal opening are

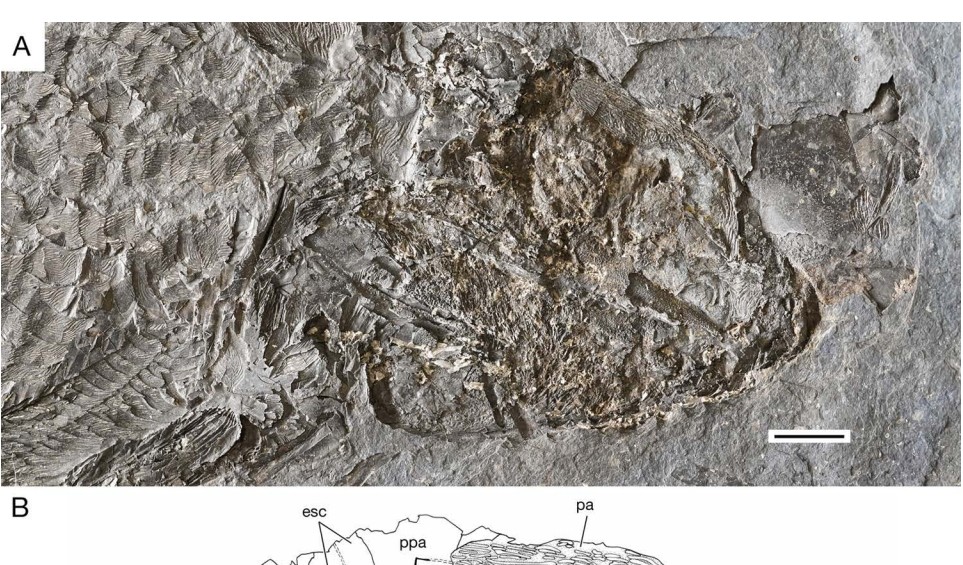

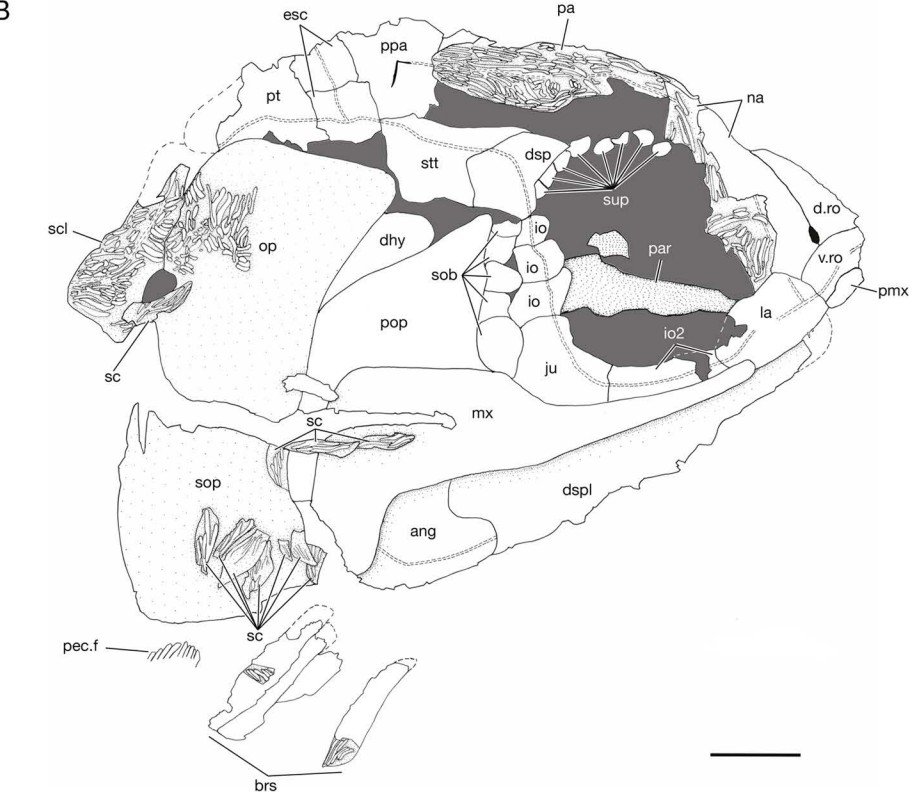

**Fig 6. Head of *Kalops loganensis* n. sp. A.** Photograph of specimen FMNH PF-2286, part; **B.** Line drawing based on specimen FMNH PF-2286, part. Abbreviations: **ang**: angular; **brs**: branchiostegal rays; **d.ro**: dorsal median rostral; **dhy**: dermohyal; **dsp**: dermosphenotic; **dspl**: dentalosplenial; **esc**: extrascapulae; **io**: infraorbitals between jugal and dermosphenotic; **io2**: infraorbital 2; **ju**: jugal; **la**: lacrimal; **mx**: maxilla; **na**: nasal; **op**: opercle; **pa**: parietal; **par**: parasphenoid; **pec.f**: pectoral fin; **pmx**: premaxilla; **pop**: preopercle; **ppa**: postparietal; **pt**: posttemporal; **sc**: scales; **scl**: supracleithrum; **sob**: suborbital; **sop**: subopercle; **stt**: supratemporotabular; **sup**: supraorbital; **v.ro**: ventral median rostral. Dashed lines represent the margins of the incomplete bones. Double dashed lines represent the sensory canals. Scale bar = 5mm.

unclear. The postparietal is a short rectangular bone. It is about 28% as long as the parietal but slightly wider than it. The postparietal sutures with the supratemporotabular laterally, and the extrascapulae posteriorly throughout dentated and/or serrated margins (*sutura dentata* type). The dorsal dermosphenotic (= intertemporal) is absent. The supratemporotabular

has an irregular shape, with its anterior portion wider than the posterior part. It has a very short suture with the lateral extrascapular posteriorly. The ventral edge of the supratemporotabular has a shallow and blunt process facing laterad but has no contact with the dorsal edge of the dermohyal. There are two pairs of extrascapulae on each side of the skull (Fig 6). They form one vertical row. All extrascapulae are square shaped, while the lateral ones are slightly larger. Both articulate with the posttemporal posteriorly and form a zigzagged margin. The medial extrascapulae also suture to each other medially throughout a *sutura dentata*. All the skull roof bones are heavily ornamented with short ridges of ganoine except the postparietal, whose ornamentation is not preserved.

The trajectory of the supraorbital canal in the parietal is bent in the orbital region. In the postorbital region, this canal is horizontal and straight in both parietal and postparietal. The supraorbital canal ends in the postparietal around its mid region. Posterior to the supraorbital canal, there is a very short anterior pit line, followed by a longer median pit line, which is still restricted to the postparietal. The posterior pit line cannot be seen in any of the specimens.

The otic canal is placed almost horizontally in the supratemporotabular, near its medial border. The posterior part of the otic canal is positioned in the lateral extrascapular. This section is short and straight, about half the length of the lateral extrascapular. The posterior section of the cephalic sensory system is tripartite with the lateral line canal and the supratemporal commissure, which is vertically straight and crosses both the lateral and medial extrascapulae.

**Neurocranium.** Unfortunately, there is no observable neurocranium preserved in the studied specimens because this area is obscured by dermal bones. An elongated and narrow element horizontally crosses the orbit in FMNH PF-2286 and is identified as the parasphenoid (Fig 6).

**Circumorbital bones.** The circumorbital ring is formed by the following bones: the nasal anteriorly, the lacrimal, the infraorbital two, and the jugal (= infraorbital three) ventrally, the three small infraorbitals and the dermosphenotic posteriorly, and the supraorbitals posterodorsally and dorsally. The sclerotic bones are not preserved in any of the specimens (Fig 6).

The lacrimal is an oval shaped bone and it is elongated on the antero-posterior axis. The posterior margin of this bone is in contact with both the infraorbital two and the maxilla. The ventral margin of the lacrimal is involved in the oral rim of the upper jaw. Two or three small conical teeth with acrodin cap are placed on the anterior portion of the ventral margin of the lacrimal. All teeth face ventrad. The infraorbital two is narrow and bar shaped. It is horizontally placed between the lacrimal and the jugal, and dorsal to the suborbital arm of the maxilla. The jugal has a nearly trapezoid shape, with its dorsal part shorter than the ventral part. There are three small irregular shaped infraorbitals placed vertically between the dorsal margin of the jugal and the ventral margin of the dermosphenotic. Their size is gradually decreasing from the ventral one to the dorsal one. The dermosphenotic is short and deep, slightly obliquely placed at the posterodorsal corner of the orbit. It has a convex posterior border articulating with the supratemporotabular and a concave anterior border. The dorsal margin of the orbit is formed by eight small rounded supraorbitals, which are arranged into one horizontal row. The first five supraorbitals are placed lateroventrally to the parietal, whereas the last three are connected to the anteroventral margin of the dermosphenotic. Posterior to the jugal and the three small infraorbitals, there is one vertical row with five small suborbitals. The dorsal most suborbital one is the smallest. It is not in contact with the dermosphenotic on its dorsal edge. The ventral most suborbital five is the biggest among them (Fig 6).

The infraorbital canal can be clearly seen on FMNH PF-2286. In general, this canal is narrow and simple, without ramifications. The canal meets with the vertical supraorbital canal, and the posterior extension of the ethmoid commissure, to form the tripartite canals in the anterior part of the lacrimal. At the midventral orbital region, the infraorbital canal is placed horizontally through the lacrimal and the infraorbital two, as well as the anterior most part of the jugal. In the jugal the infraorbital canal bends dorsad. The infraorbital canal is vertical across the three infraorbitals between jugal and dermosphenotic. In the dermosphenotic, the infraorbital canal connects with the otic canal near the posterodorsal margin of this bone.

**Upper jaw.** The upper jaw is formed by the premaxilla, the lacrimal, and the maxilla. In general, the oral rim of the upper jaw is longer than the lower jaw, bearing an anteroventral notch at the lateral profile of the skull (Fig 6).

The maxilla is the largest bone in the upper jaw. It has two distinct sections: the suborbital arm and the posterior plate. The maximum length/depth ratio of this bone is almost 2.3:1. The narrow, long and straight suborbital arm contributes to

the anterior half of the maxilla, its depth is gradually increasing from anterior to posterior. The anterior tip of the suborbital arm is ventral to the orbital ring. Its depth gradually increases from anterior to posterior, and the highest point of its dorsal margin is lower than the midline of the orbital ring. The posteroventral process on the posterior plate is short and deep, about half as depth as the posterior plate. On the oral margin of the maxilla, from its anterior most part until the beginning of the posteroventral process, there is one row of sharp, straight, and narrow conical teeth with acrodin cap. The posterior plate of the maxilla is strongly ornamented by continuous short ridges of ganoine. These ganoid ridges have different orientations. The ridges near the oral margin of the maxilla are almost horizontal, the ridges near the dorsal margin of the maxilla are diagonal, posterodorso-anteroventrally placed, and the ridges on the posteroventral process are nearly vertical. However, the ornamentation of the suborbital arm is unclear (Figs 6–11). The medial surface of the maxilla is almost flat and smooth. The horizontal longitudinal lamina is preserved in the medial surface as a deep groove, which is horizontally across the suborbital arm and the ventral part of the posterior plate. The horizontal longitudinal lamina is ended slightly anterior than the origin of the posteroventral process.

**Lower jaw and suspensorium**. The lower jaw consists of a dentalosplenial (= dentary) and the angular. The lateral preservation of the specimens studied prevents any further observation of the elements in medial side (Figs 6, 9, and 10).

The dentalosplenial is elongated, over half of the skull length. The depth of the dentalosplenial gradually increases from anterior to posterior. The oral margin and the teeth condition are unknown because these parts are covered by the upper jaw in all specimens. Sutured posteriorly to the dentalosplenial, there is the angular, which is about ¼ as long as the dentalosplenial. Their suture is wavy and "s" shaped. These two bones are strongly ornamented with obliquely oriented long ridges of ganoine. In all specimens the oral margin of the lower jaw is covered by the upper jaw, preventing any observation of the dentition.

The mandibular canal can be only seen in its posterior part, restricted to the angular. This shortly exposed section is straight on its anterior part and anteroventro-posterodorsally oblique at its posterior part.

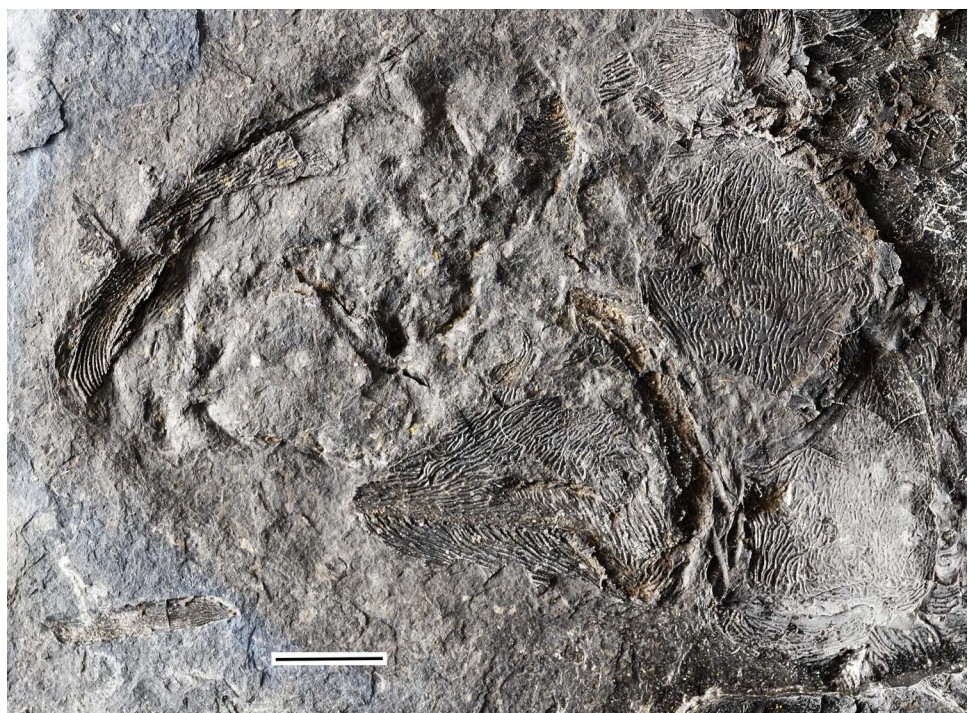

**Fig 7. *Kalops loganensis* n. sp.** Skull photograph of specimen FMNH PF-2286, counterpart. Scale bar = 5mm.

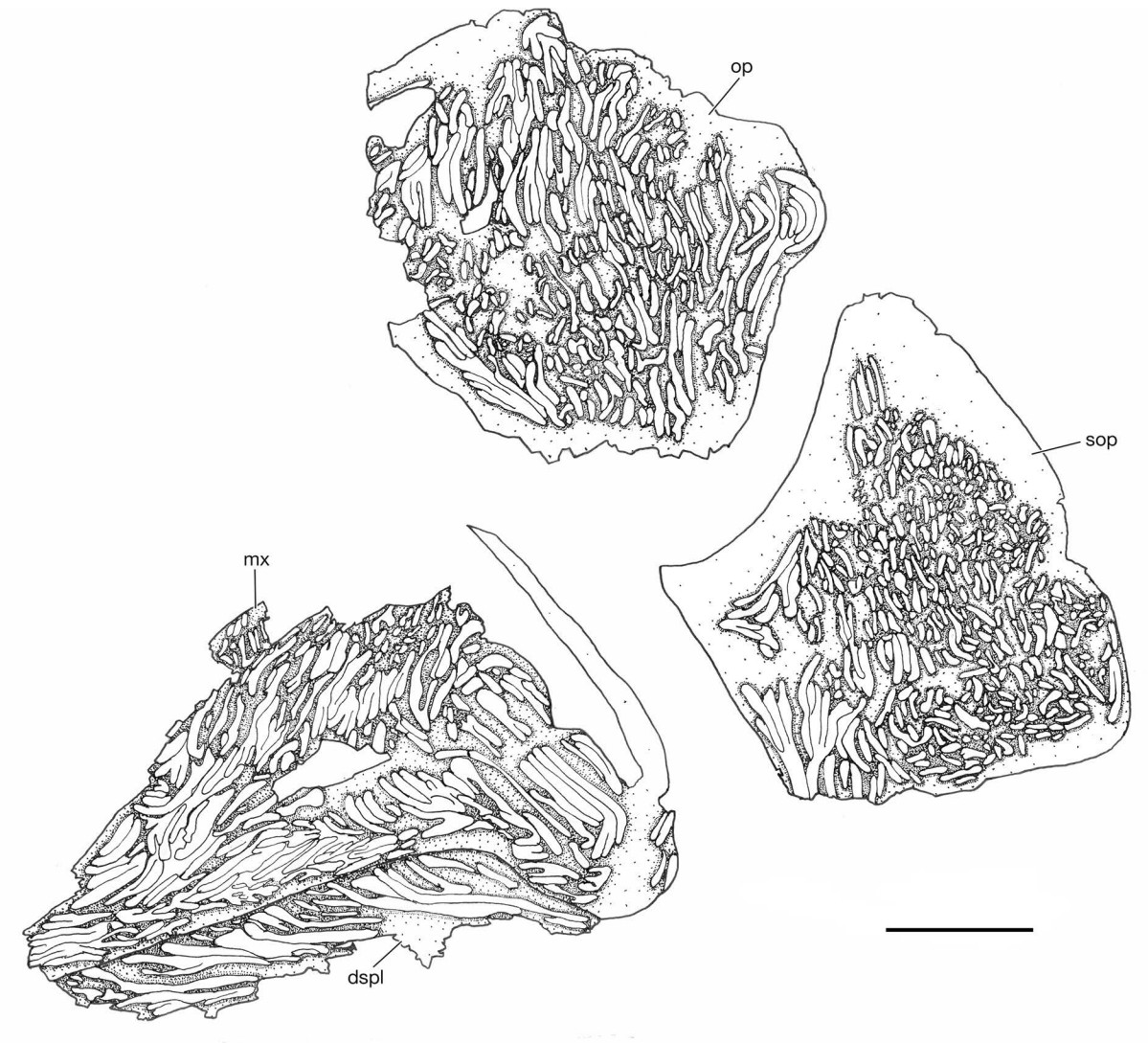

**Fig 8. *Kalops loganensis* n. sp. Skull bones, line drawing based on specimen FMNH PF-2286, counterpart.** This drawing only includes the large identifiable elements. Note the strong ornamentation on these bones. Abbreviations: **dspl**: dentalosplenial; **mx**: maxilla; **op**: opercle; **sop**: subopercle. Scale bar = 5mm.

**Opercular bones, branchiostegal rays, and gular plates**. The bones described in this section include the preopercle, the opercle, the subopercle, the dermohyal, and the branchiostegal rays. The gular plates are not preserved (Figs 6, 7, 9, and 10).

The preopercle has a long and broad anterodorsal branch and a narrow and short ventral branch. The angle between these two branches is 90degrees. The preopercle does not contact the dermosphenotic, or any of the skull roof bones. The antero-dorsal branch is large, over half as deep as the orbital depth and as long as the length of the posterior plate of the maxilla. The anterior margin of the anterodorsal branch is vertical and straight. The ventral branch is about half as deep as the posterior plate of the maxilla. Its ventral margin is nearly at the same horizon of the oral rim of the upper jaw. A relatively large and splint like dermohyal is located between the posterodorsal margin of the preopercle and the anterior margin of the opercle.

The opercle is most complete in FMNH PF-2286, and description is based on PF-2286 (Fig 6). This bone is trape-zoid and deep, having a depth/length ratio of 1.5. The ventral margin is slightly longer than the dorsal margin. This bone

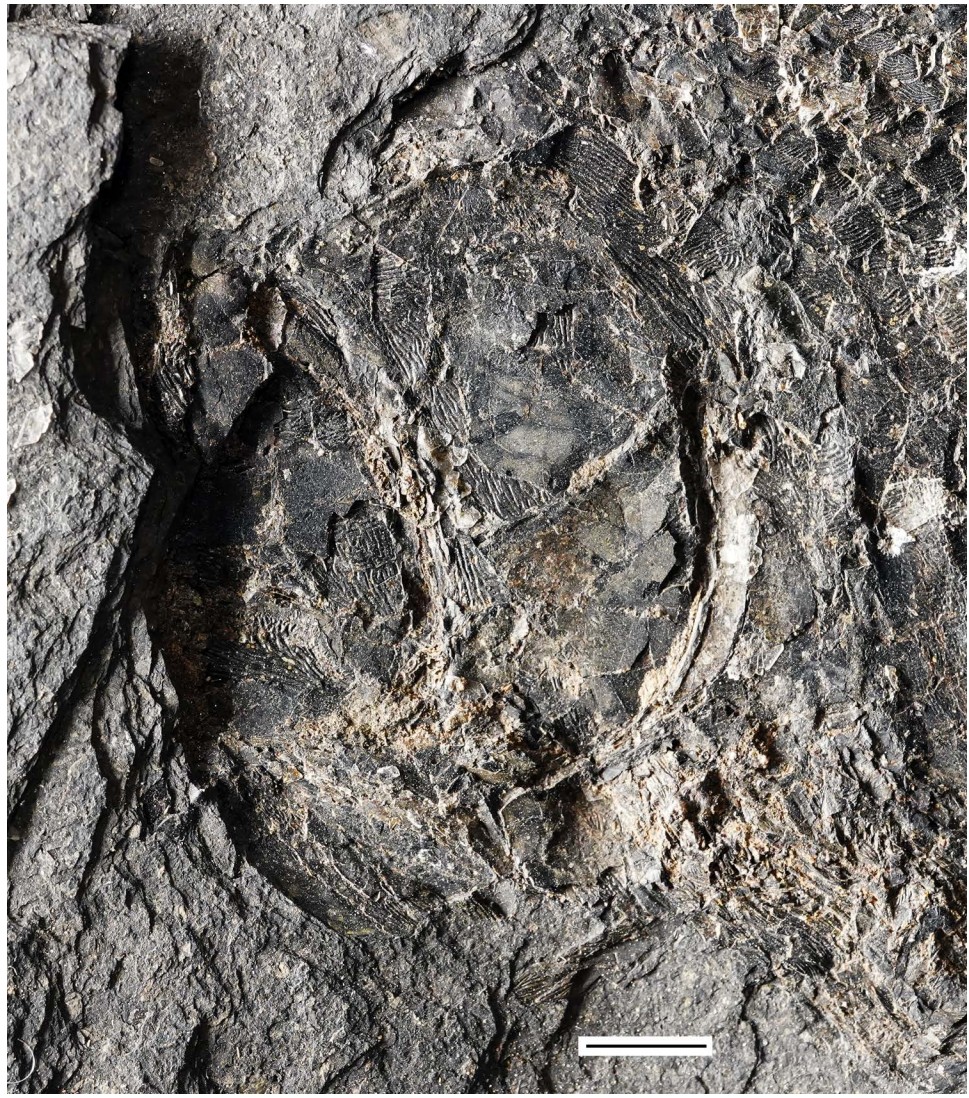

**Fig 9. *Kalops loganensis* n. sp. Skull photograph of specimen FMNH PF-5919.** Scale bar = 5mm.

articulates with the posttemporal posterodorsally, the supracleithrum posteriorly, and the subopercle ventrally. The dorsal margin of the opercle is higher than the dorsal margin of the dermohyal. The opercle has concave anterior and ventral margins, and a slightly convex posterior margin. The subopercle is wider and shallower than the opercle. In general, the size of the subopercle is over half than that of the opercle. The posterior margin of the subopercle is longer than the anterior margin, being its dorsal margin an oblique line, which is also concave. The subopercle in the holotype has an anteroventral process but lacks an anterodorsal process.

The branchiostegal rays are preserved better in FMNH PF-5919 with nine branchiostegal rays extending from the ventral part of the subopercle to the posteroventral part of the lower jaw. The first branchiostegal ray is the largest ray among the nine, and it is oval shaped, with the anterior part larger than the posterior part. Other branchiostegal rays are elongated and rectangular shaped.

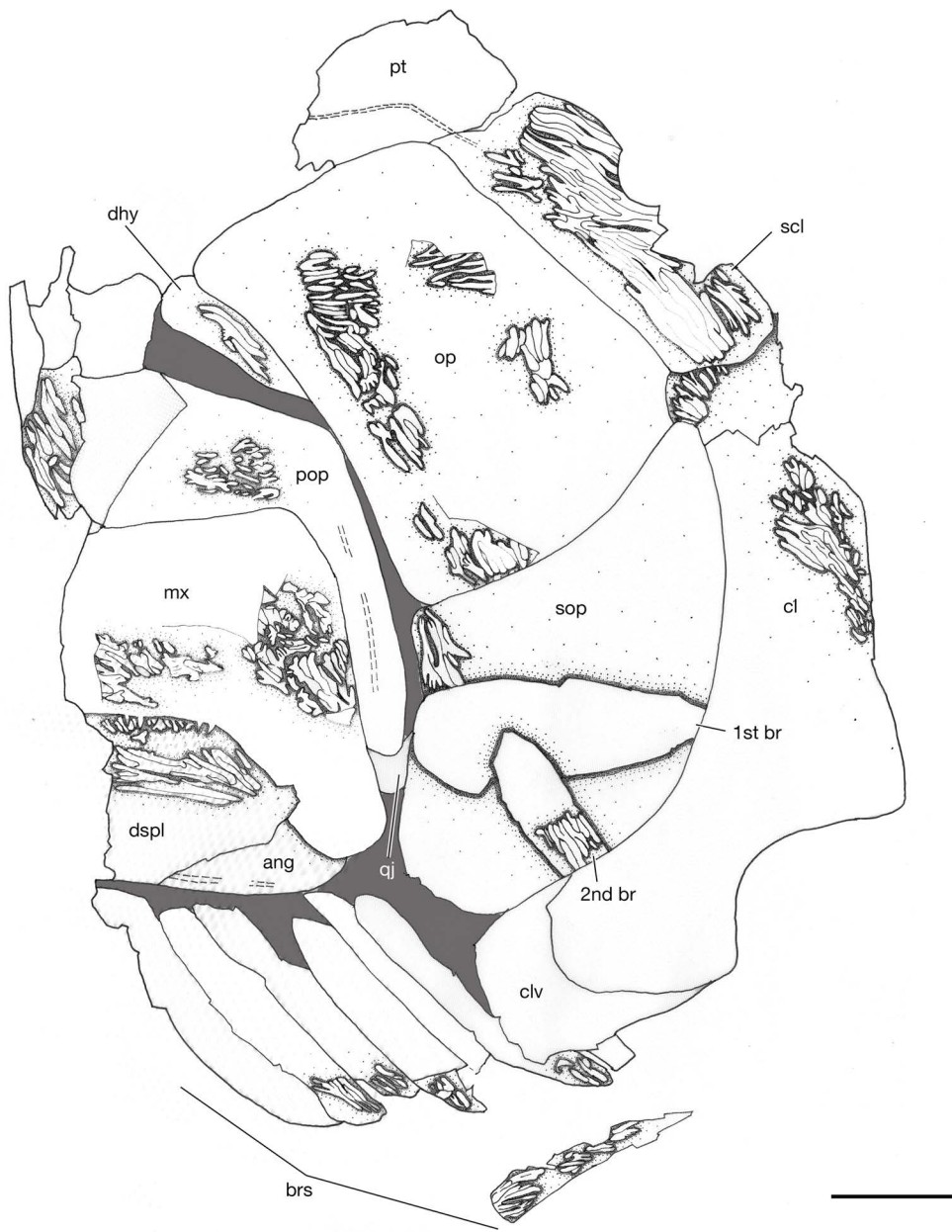

**Fig 10.** *Kalops loganensis* n. sp. Skull line drawing based on specimen FMNH PF-5919. Abbreviations: **1st br**: first branchiostegal ray; **2nd br**: second branchiostegal ray; **ang**: angular; **brs**: branchiostegal rays; **cl**: cleithrum; **clv**: clavicle; **dhy**: dermohyal; **dspl**: dentalosplenial; **mx**: maxilla; **op**: opercle; **pop**: preopercle; **pt**: posttemporal; **qj**: quadratojugal; **scl**: supracleithrum; **sop**: subopercle. Scale bar = 5mm.

All bones described above are ornamented with tubercles and short ridges of ganoine. The preopercle only has ornamentation preserved in a small area on the anterodorsal branch in FMNH PF-5919 (Figs 9 and 10). The ornamentation consists of short ridges and some small tubercles. The dermohyal is ornamented by several ridges of ganoine that are placed parallel to the longitudinal axis of this bone (Figs 9 and 10). The opercle and subopercle are heavily ornamented by short ridges and tubercles of ganoine (Figs 7 and 8). None of the branchiostegal rays preserve the complete

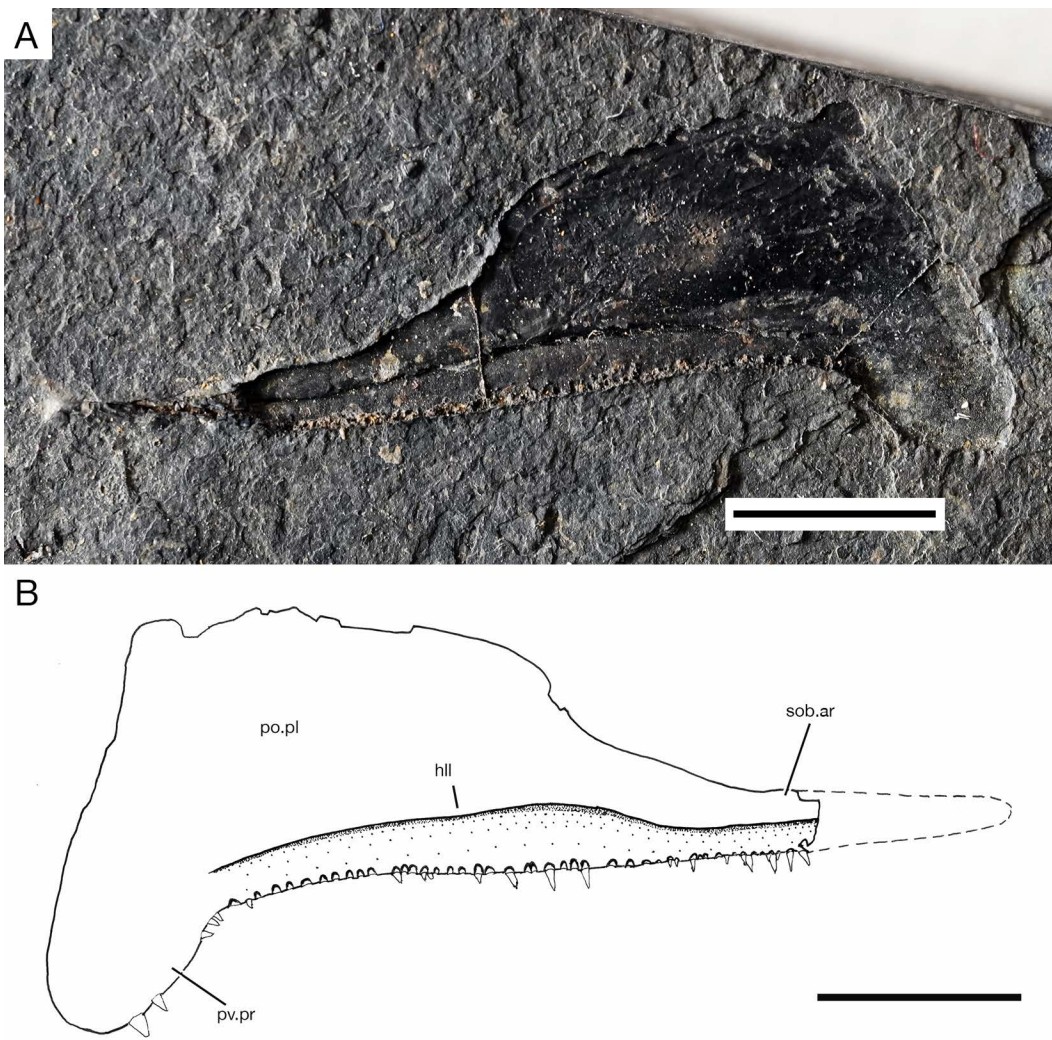

**Fig 11.** *Kalops loganensis* **n. sp. A separate maxilla on specimen FMNH PF-2286, in medial view. A.** Photograph, PF-2286 part; **B.** Line drawing based on specimen FMNH PF-2286, counterpart. Abbreviations: **hll**: horizontal longitudinal lamina; **po.pl**: posterior plate; **pv.pr**: posteroventral process; **sob.ar**: suborbital arm. Scale bar = 5mm.

ornamentation. Most of the preserved ornamentation are on the anterior or posterior ends of the rays, and consist of dense short ridges of ganoine. These ridges are parallel to the longitudinal axis of the branchiostegal rays.

The preopercular canal is only preserved as a short section in the ventral branch of the preopercle in FMNH PF-5919.

**Pectoral girdle and fin**. The following bones forming the pectoral girdle are preserved: posttemporal, supracleithrum, cleithrum, and clavicle. Although that the posttemporal is included here, this bone is the element connecting the pectoral girdle with the cranium.

The posttemporal is a nearly triangular bone with an antero-posterior elongation. This bone articulates with the lateral and medial extrascapulae anteriorly, and the supracleithrum posteroventrally (Figs 6, 9, and 10). *K. loganensis* n. sp. does not have presupracleithrum.

The supracleithrum is rectangular shaped, narrow, deep, and obliquely placed. It is shallower than the opercle and articulates with the opercle anteriorly. Ventrally the supracleithrum covers the dorsal most part of the cleithrum (Figs 6, 9, and 10).

The cleithrum is incomplete in FMNH PF-5919, but there is a disarticulated cleithrum in FMNH PF-2286, which is completely preserved (Fig 12). This bone is deep and narrow, with a dorsal and a ventral section. The dorsal section occupies over 2/3 of the depth of the bone. It is almost triangular shaped, having a pointed dorsal tip, which is also bending anteriad. The ventral portion of the dorsal section bends anteriad as well and connects to the ventral section of this bone. The ventral section of the cleithrum is rectangular, about the same length as the dorsal section. The cleithrum has a convex anteroventral margin and a 90-degree wide posteroventral notch, where the pectoral fin is attached here (Fig 12). The clavicle has only its posterior most part preserved in FMNH PF-5919. That part is articulated with the cleithrum anteroventrally. Among the dermal bones of the girdle, only the supracleithrum and the cleithrum are heavily ornamented with short and long ridges of ganoine.

The pectoral fin is preserved *in situ* in FMNH PF-5919, but the condition is too poor to get any details of its structure. Fortunately, in the FMNH PF-2286 there is a relatively more complete external mold of the pectoral fin. This fin is placed near the ventral margin of the fish. There is no lobe structure attached to its base. In general, the fin is short, only 12% of the specimen's length, which means the value of the pectoral fin length/standard length ratio is even lower. It has 13 or 14

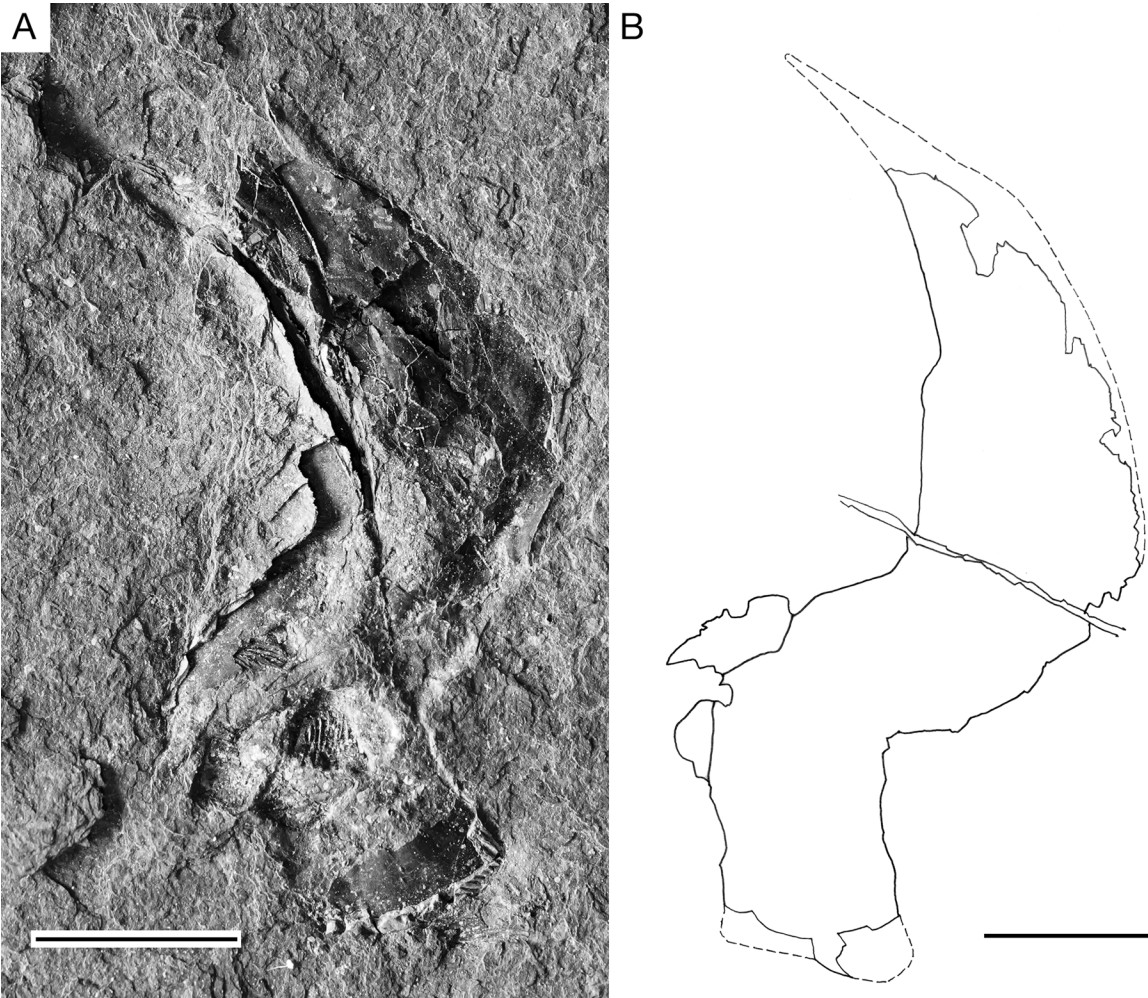

A

B

**Fig 12. *Kalops loganensis* n. sp. An isolated cleithrum on specimen FMNH PF-2286. A.** Photograph, PF-2286, part; **B.** Line drawing based on PF-2286, part. Scale bar =5 mm.

fin rays. All are segmented from proximal to distal. The marginal rays are longer than the last few rays, giving a triangular lateral profile to the pectoral fin. The condition of the basal fulcra and fringing fulcra is unknown.

**Pelvic girdle and fin**. There is no pelvic girdle observable in the studied specimens because that area is obscured by the scales. The pelvic fin is poorly preserved in FMNH PF-5919. It originates at the 10th vertical scale row. Only three fin rays are preserved, which are segmented proximo-distally.

**Median fins**. The anal fin is not preserved on all specimens. The dorsal fin in FMNH PF-5919 has a relatively better preservation, although it is still incomplete. The dorsal fin originates at the 20th vertical scale row in the FMNH PF-2286 and at the 23rd vertical scale row in FMNH PF-5919. Its lateral profile is probably triangular shaped. The fin has one small basal fulcrum and followed by 13 or 14 rays segmented from proximal to distal. Fringing fulcra are unknown on the dorsal fin.

The caudal fin is lost on FMNH PF-5919, but the hypaxial lobe is preserved partly on FMNH PF-2286. The hypaxial lobe has two basal fulcra, and 29 rays. Among them, seven rays are segmented but not branched thus they are confirmed as procurrent rays. The most distal lepidotrichial segments are spine-like and longer than the proximal segments. Small fringing fulcra appear on the ventral margin of the marginal ray segments of the procurrent rays, between the most distal segments. The fringing fulcra are rhomboid shaped and swollen distally. Each marginal ray segment aligns with one fringing fulcrum. The epaxial lobe is only preserved as five long basal fulcra.

**Squamation**. The regions of the scales described here are following the criteria of Esin [45] (see Fig 4). The body is fully covered with rhomboid scales with ganoine (Figs 13 and 14). There are approximately 40 vertical scale rows between the skull and the hinge line. The mid-flank scales of the first seven vertical rows close to the skull are trapezoid and deepened (Figs 13A and 14A). Their ventral margin is longer than the dorsal margin. These scales also have a maximum depth/length ratio of 1.5. Starting from the first vertical row, this ratio of the deepened scales decreases, until the eighth row, where the ratio reaches 0.7. The deepened scales also have the peg-and-socked articulation and the anterodorsal process. The ornamentation consists of long and short ridges of ganoine. Most of them are anterodorso-posteroventrally oriented. The mid-flank scales from the eighth vertical row to the 14th row are nearly rectangular shaped but have their anterior part bends dorsad (Figs 13B and 14B). Their posterior margin is deeper than the anterior margin. These scales also have the peg-and-socket articulation. The ornamentation is formed by long ridges (over half of the scale's length) of ganoine. These ridges are tightly arranged with an anterodorso-posterovental orientation. Starting with the 15th vertical row, the mid-flank scales are elongated rhomboid shaped and without peg-and-socket articulation (Figs 13C and 14C). They have the same ornamentation pattern as the scales from the eighth to the 14th row. But there are small brushes like crenulations at the anterior end of the ridges. The scales near the dorsal margin of the body and anterior to the dorsal fin are triangular shaped without peg-and-socket articulation (Figs 13E and 14E). Their ornamentation consists of long ridges tightly arranged and oriented anterodorso-posteroventrally. The four horizontal rows of the scales near the dorsal margin of the body trunk, posterior to the origin of the dorsal fin (Figs 13G and 14G), and the five horizontal rows of the scales near the ventral margin of the body (Fig 13F), are highly elongated, with a depth/length ratio as low as 0.17. They do not have the peg-and-socket articulation but have an anterodorsal process. The ornamentation of these scales is the same as the scales on the same vertical row.

The dorsal and ventral scutes have various morphologies. The dorsal scutes anterior and posterior to the dorsal fin are rhomboid and have an antero-posterior elongation (Figs 13I, 13II and 14I, 14II). Their size is nearly the same as the scales ventral to them. The ornamentation consists of long ridges of ganoine tightly arranged, and oriented antero-posteriorly. The ventral scutes at the anterior part of the body, probably anterior to the position of the anal fin, are gigantic, over five times as large as the dorsal scutes. The only complete one shows an oval shape with the antero-posterior elongation, having its anterior end broader than the posterior one (Figs 13III and 15A). The ornamentation consists of highly dense long ridges of ganoine tightly arranged. There is one mid ridge from the posterior end of the scute until the mid-region. Almost all the other ridges are the branches of the mid ridge and parallel to the rounded lateral margins of the scutes. On the anterior tips of these ridges there are several short parallel grooves, giving fine crenulations on the anterior

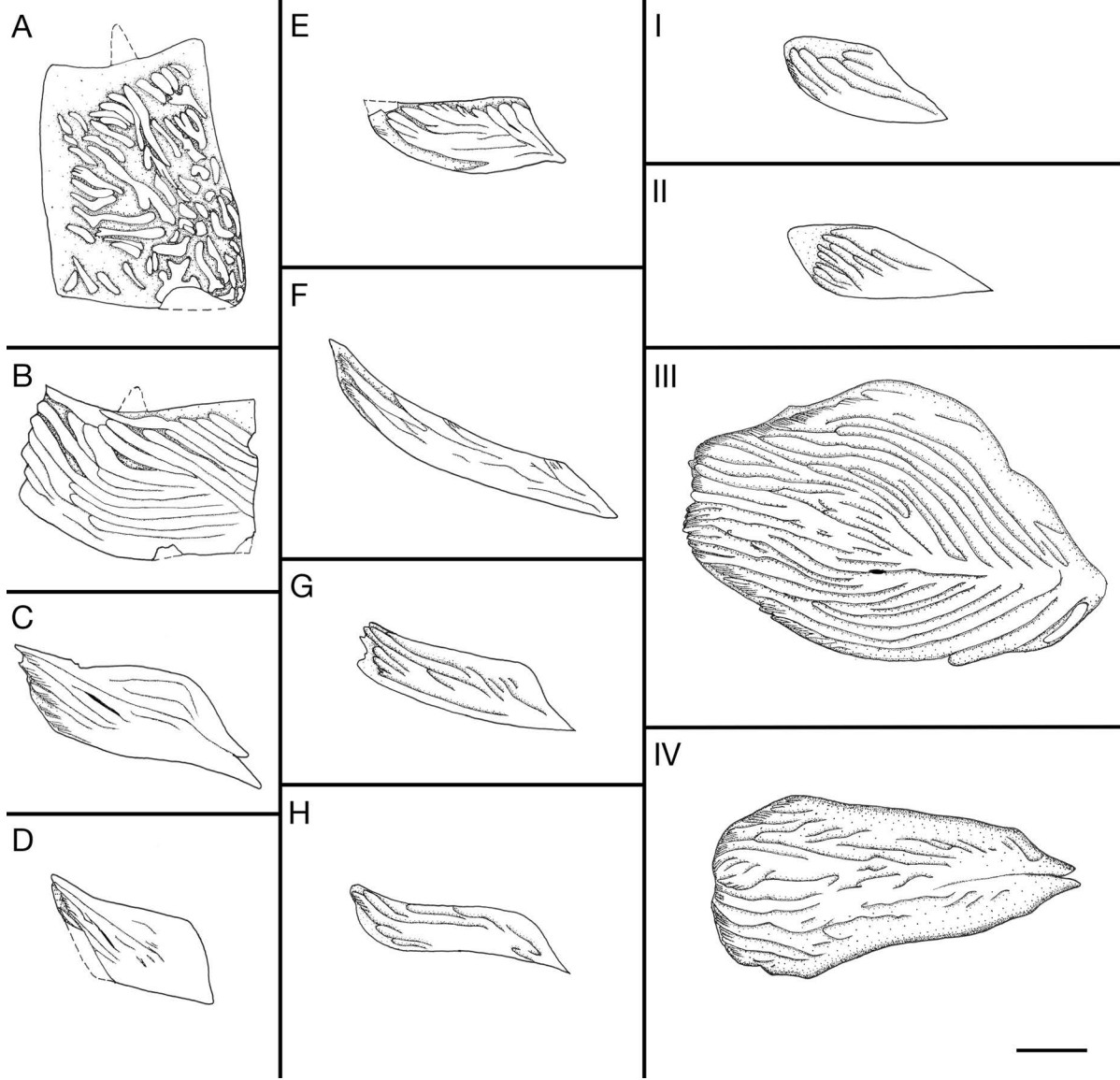

**Fig 13. *Kalops loganensis* n. sp. Line drawings of the scales from specimen FMNH PF-5919.** The letters are marking the regions in Fig 5. Scale bar = 1mm.

end of these ridges. The ventral scutes between the anal fin and the caudal fin are slightly smaller than the ventral scutes described above. Their shape is like an elongated water drop, with a broader anterior part and narrower posterior end (Figs 13IV and 15B). The posterior tip is bifurcated. The ornamentation is a combination of short grooves on the posterior half, and the dense long ridges of ganoine on the anterior half. The anterior tips of these ridges are also finely crenulated.

The trajectory of the lateral line on the body trunk is unclear. There is no scale showing lateral line openings or pit organ pores.

## Phylogenetic results

The parsimony phylogenetic analysis recovered 1351 most parsimonious trees. The topology of the strict consensus tree is shown in Fig 16. The tree length is 1351, the consistency index (CI) is 0.2184, and the retention index (RI) is 0.6450.

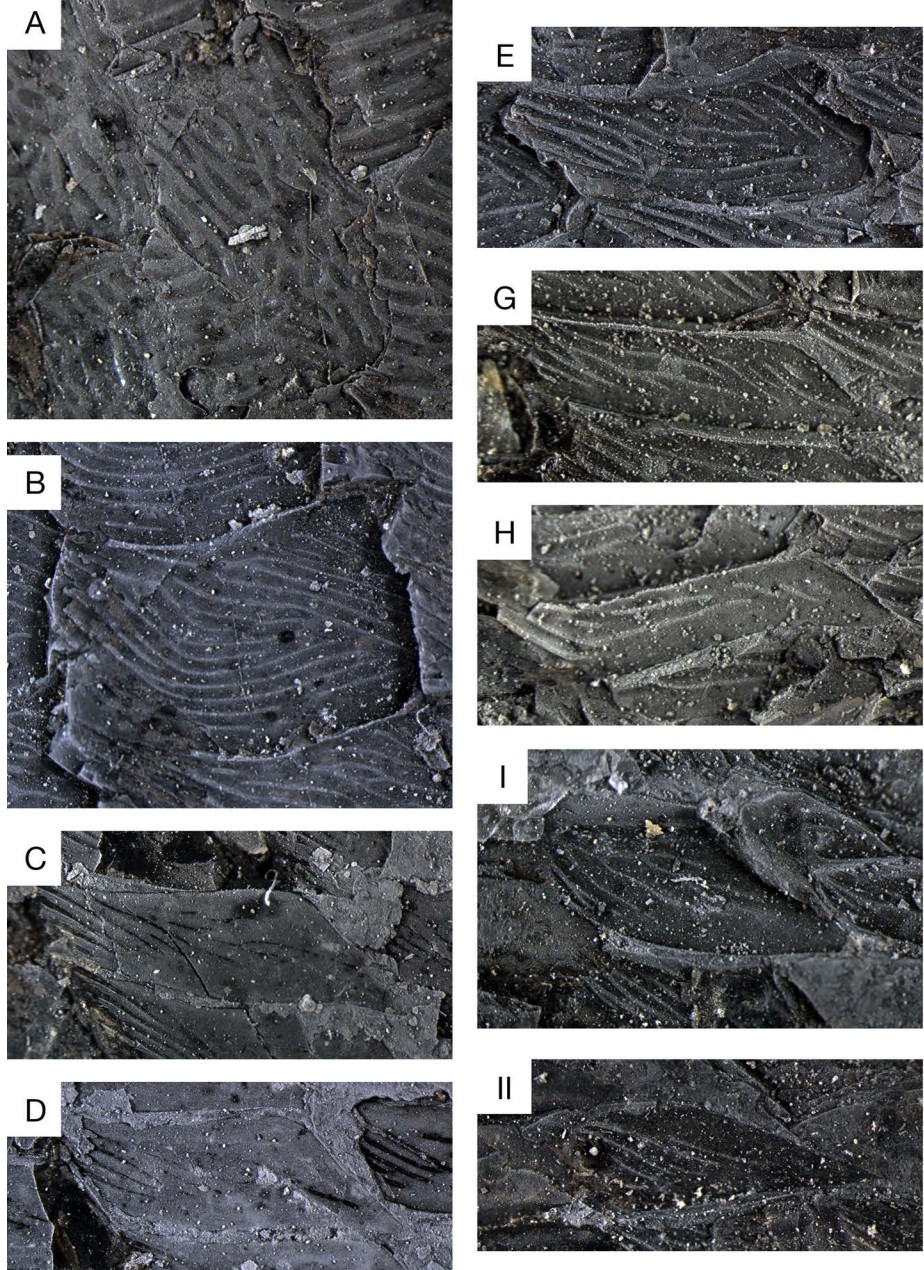

**Fig 14. Kalops loganensis n. sp. Photographs of the scales in specimen FMNH PF-5919.** The letters are marking the regions in Fig 5. Note that the size of the scales is not under the same enlargement.

The result of this phylogenetic analysis indicates that *Kalops loganensis* n. sp. is a member of *Kalops* (Fig 16, node B), and stands as the sister taxon of the two Bear Gulch *Kalops* species (Fig 16, node C). The *Kalops* node is supported by nine synapomorphies, which all are homoplastic: premaxilla is confined to the region anterior to the orbit (Ch. 6 [1]); parietal (= frontal) is shorter than postparietal (parietal) (Ch. 28 [0]); parietals (= frontals) broad posteriorly and tapering anteriorly (Ch. 29 [0]); the infraorbital canal in jugal does not have ramification (Ch. 55 [0]); three or more supraorbitals (Ch. 58

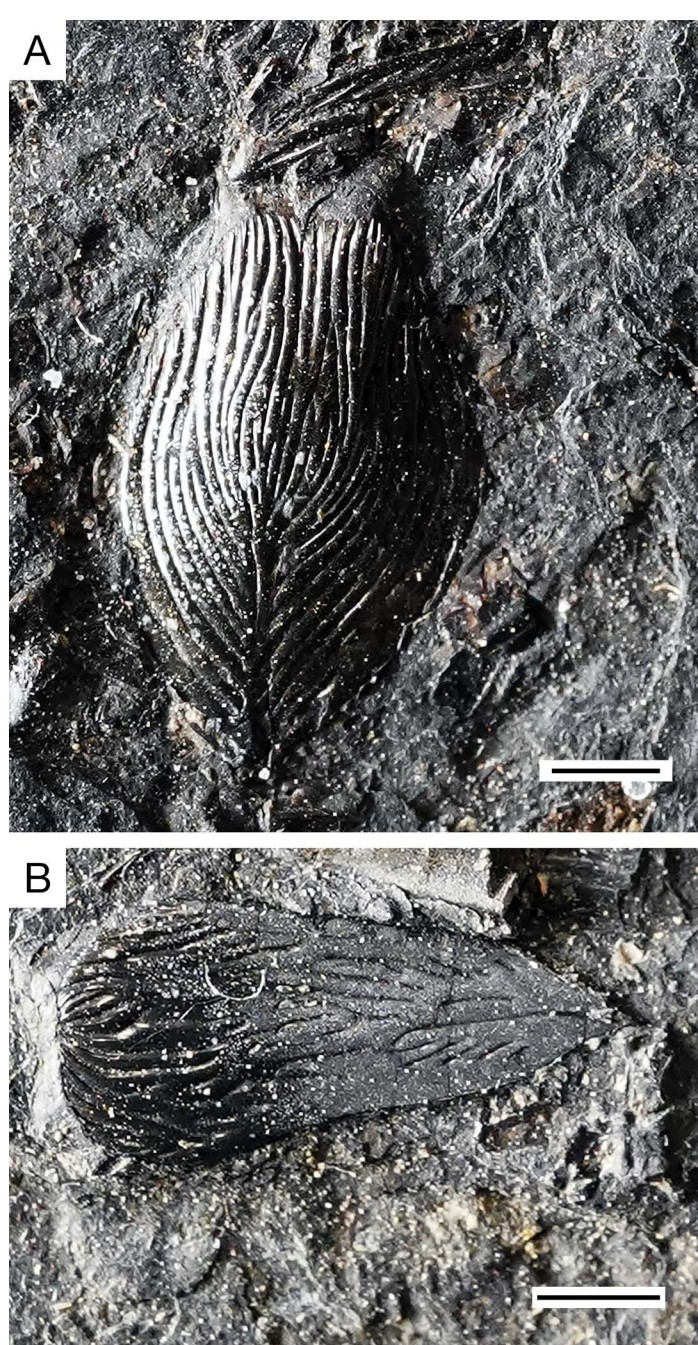

**Fig 15. *Kalops loganensis* n. sp. Photographs of the scutes on specimen FMNH PF-5919. A.** Same scute in Fig 13III; **B.** Same scute in Fig 13IV. Scale bar =1mm.

[2]); the opercle is at least twice as high as the subopercle (Ch. 110 [0]); subopercle has anteroventral process (Ch. 113 [1]); opercular process is absent (Ch. 211 [0]); pectoral fin endoskeleton extends far beyond body wall (fins lobate) (Ch. 238 [0]). The clade of *K. monophrys* and *K. diophrys* is supported by two homoplastic features: one pair of extrascapulae (Ch. 43 [0]); a single median extrascapular is present (Ch. 45 [0]). The only autapomorphy of *K. loganensis* n. sp. is the

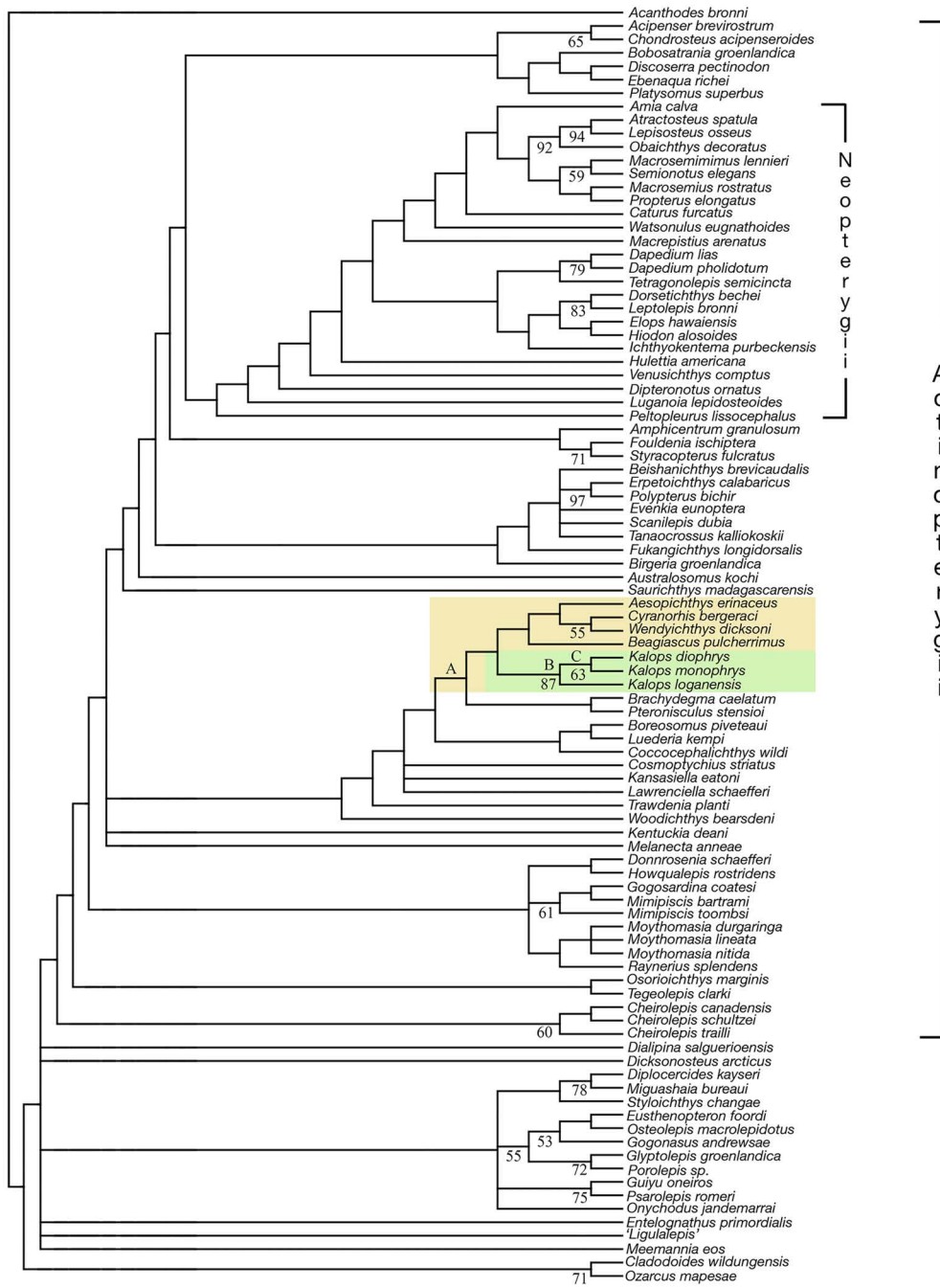

**Fig 16. Phylogenetic relationships of *Kalops loganensis* n. sp. and its relatives.** The data matrix is based on 95 taxa and 265 characters, and *Acanthodes bronni* is selected as the outgroup. Strict consensus tree of 1351 most parsimonious trees. Tree length is 1351, consistency index (CI) = 0.2184, retention index (RI) = 0.6450. Numbers under the nodes are the bootstrap values higher than 50%. The *Kalops* clade is highlighted in green, and its relatives are highlighted in yellow.

absence of presupracleithrum (Ch. 226 [0]). The *Kalops* clade is nested within the monophyletic group that most of the members, except *K. loganensis* n. sp., are from the Serpukhovian Bear Gulch Limestone (Fig 16. node A). This node is supported by two homoplastic characters: the mandibular canal traces the ventral margin of the lower jaw along the entire length (Ch. 76 [0]); one infradentary (angular only) (Ch. 89 [2]). For the characters states of other nodes, see Supplementary Material 1.

## Discussion

**Comparison with the other two *Kalops* species**. *Kalops loganensis* n. sp. appears to be the largest kalopid fish, longer than the 116mm *K. monophrys* and 96mm *K. diophrys*. Poplin & Lund [13] listed several features that distinguish *Kalops* from other early actinopterygians, including short dermosphenotic, nasal does not contact to dermosphenotic, five to six infraorbitals, subterminal mouth, and lobed pectoral fin. Poplin & Lund [3] also emphasized three features that are the most obvious ones, including small numerous supraorbitals, small numerous suborbitals, and an accessory row of extrascapulae.

K. loganensis n. sp. shares almost all features with other *Kalops* except the pectoral fin lobe, which remains unknown due to the poor preservation, and the accessory extrascapulae row, which is only present in *K. diophrys* and absent in both *K. monophrys* and *K. loganensis* n. sp.

The dermosphenotic in *K. loganensis* n. sp. is short compared to the surrounding skull roof bones, more similar to *K. monophrys*, while *K. diophrys* has a longer dermosphenotic.

All the three *Kalops* have lacrimal, infraorbital 2, and jugal bones. But the conditions between the jugal and dermosphenotic are different. *K. monophrys* and *K. loganensis* n. sp. have three infraorbitals in this region, while *K. diophrys* has four. *Kalops loganensis* n. sp. has the fewest number of supraorbitals, eight forming one row, *K. monophrys* has 13 supraorbitals in one row, and *K. diophrys* has 22 in two rows, 12 in the upper row, and ten in the lower row.

The suborbitals are different in all species. *Kalops. loganensis* n. sp. has five suborbitals, placed in a vertical row, with reduced size from ventral to dorsal. The ventral most suborbital is two times larger than the dorsal most one. *K. monophrys* has five suborbitals but they are oriented in an obliquely vertical row. The ventral most suborbital is more than ten times larger than the dorsal most suborbital. *K. diophrys* has the most numerous suborbitals, 12 of them organized in three vertical rows. The largest suborbital is still the most ventral one, nearly ten times larger than the smallest suborbital bone.

Another feature that is different across all species is the extrascapulae. A median extrascapular appears in both species from the Bear Gulch but is absent in *K. loganensis* n. sp. *K. loganensis* n. sp. has two pairs of lateral extrascapulae and no accessory extrascapulae. *K. monophrys* and *K. diophrys* have one pair of lateral extrascapulae. In addition, *K. diophrys* has one vertical row of five small accessory extrascapulae, anterior to the lateral extrascapular. One specimen of *K. monophrys*, Carnegie Museum CM35395, preserved one small, rounded bone surrounded by lateral extrascapular, postparietal, and supratemporotabular, and it was identified by Poplin & Lund [13] as an accessory extrascapular. However, this feature was only present on one specimen [13, fig.2-2] without any description. Thus, this specimen needs further examination.

**Phylogenetic assessment**. Across different earlier analyses, the phylogenetic position of *Kalops* drifts within the stem Actinopterygii group. It was recovered as the sister group of other Bear Gulch taxa such as *Aesopichthys* in recent studies [49–51], or as the sister group of a larger Bear Gulch clade including *Aesopichthys*, *Proceramala*, *Wendyichthys*, and *Cyranorhis* by Cloutier & Arratia [52, fig. 10A]. Argyriou et al. [53] recovered a clade including *Kalops*, *Beagiascus*, *Coccocephalus*, *Boreosomus*, and *Luderia*, and this clade is sister to the *Pteronisculus* + *Aesopichthys* + *Cyranorhis* + *Wendyichthys* clade. Some other studies suggested that the Bear Gulch actinopterygian species are not closely related to each other. *Kalops* clade is the sister group of *Elonichthys* + *Palaeoniscum* clade in Mickle et al. [54]. However, Mickle [38] hypothesized that *Kalops* is closely related to *Gonatodus*. Both are nested in a clade without any other members from Bear Gulch [39, fig. 9, node 46]. Xu

et al. [55] proposed that *Kalops* is in a pectinated paraphyletic position, between *Boreosomus* and *Palaeoniscum*. In the phylogenetic result from Stack & Gottfried [56], *Kalops* is in a polyphyletic position, along with some other early actinopterygians. In summary, although using different sets of characters, the hypotheses of relationships recovered in the analysis presented here are more in agreement with those of Cloutier & Arratia [52] than other studies, despite the fact that the parameters, such as the selection of taxa and characters, and the outgroup taxon used in both works are different.

**Survivability**. The new discovery of *Kalops loganensis* n. sp. extends the youngest age of this genus to nearly 13 million years from Serpukhovian to Moscovian [6,57–59]. Geographically, The Bear Gulch Limestone is part of the Williston Basin, and the Logan Quarry Shale is on the east side of the Illinois Basin [19,60]. During the Mississippian these two basins were separated shallow marine systems [61, fig. 1]. The two shallow seas were connected as the subunits of the North American Midcontinent Sea (NAMS) during the Mid-Pennsylvanian stage. The Williston Basin formed the northeastern boundary of the NAMS, and the Illinois Basin, which is more than 500 km south of the Williston Basin, formed part of the southeastern boundary [15,62].

In the NAMS, the repeated stratigraphically similar layers (transgression limestone, gray/black shales, regression limestone) indicate the cycles of eustatic fluctuations across the three North American Pennsylvanian regional stages, from Desmoinesian, Missurian to Virgilian. These fluctuations were driven by the global ice age intervals during the Late Paleozoic and were defined as "cyclothems" [57–59,63]. So far, all the cyclothems are studied and named based on local geological units. The Logan Quarry Shale member is in the Upper Tiawah cyclothem, which is the 22nd cyclothem [57,64].

**Potential habitat**. *Kalops* has a fusiform body shape, together with the location of the fins, indicating that this genus is a fast swimmer around the epipelagic of the tropical shallow marine [65]. Similar lifestyles can be found in the Triassic *Pteronisculus* [66]. This ecological niche of adapting to the epipelagic marine might explain how this fish survived through the cyclothems. They were able to follow the rise and retreat of the sea level, distributed from Williston Basin to Illinois Basin, and tolerate the cyclothems in the NAMS.

## Conclusions

Studies of two specimens from the Logan Quarry reveal a new species of *Kalops*, *K. loganensis* n. sp. as the youngest representative of its genus in the Moscovian of North America. The morphological study and phylogenetic analysis confirm that the new species is the sister of the other two known *Kalops* species. The genus is interpreted as a stem Actinopterygii, closely related to other Bear Gulch fishes with a fusiform body, such as *Beagiascus*, *Cyranorhis*, and *Wendyichthys*. Based on geological evidence, *Kalops* is the only actinopterygian genus occurring in both the Bear Gulch Limestone and a younger unit, Logan Quarry. Further studies can focus on the functional morphomatrix analysis to quantify the size and shape of *K. loganensis* n. sp. in regard to survivorship. Another direction is the faunal dynamics analysis of how the cyclothems affect the marine vertebrate faunas in the NAMS.

No permits were required for the described study, which complied with all relevant regulations.

## Supporting information

**S1 File. List of characters used in the phylogenetic analysis and the characters states that support the nodes of the phylogenetic result.**
(DOCX)

**S2 File. Data matrix for the phylogenetic analysis in NEXUS format.**
(NEX)

**S1 Fig. Tif.** XXX.
(TIF)

## Acknowledgments

Greatly thanks to H.-P. Schultze and G. Arratia for the comments and suggestions on this manuscript, and J. Chorn for the suggestions on photographing. I also appreciate A. Stroup and W. Simpson for accessibility to the Field Museum specimens. My thanks to the editor, and the two anonymous reviewers for their constructive comments on this manuscript.

## Author contributions

**Formal analysis:** Chenchen Shen.

**Funding acquisition:** Chenchen Shen.

**Investigation:** Chenchen Shen.

**Methodology:** Chenchen Shen.

**Writing – original draft:** Chenchen Shen.

**Writing – review & editing:** Chenchen Shen.

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
