## [Decision Letter · Decision Letter 0]

24 Sep 2024

PONE-D-24-33140A new early Actinopterygii from the Mid-Pennsylvanian Logan Quarry Shale member of IndianaPLOS ONE

Dear Dr. Shen,

Thank you for submitting your manuscript to PLOS ONE. After careful consideration, we feel that it has merit but does not fully meet PLOS ONE’s publication criteria as it currently stands. Therefore, we invite you to submit a revised version of the manuscript that addresses the points raised during the review process.

**ACADEMIC EDITOR: The reviewers find the manuscript interesting but evidenced several issues that must be addressed in order to make your manuscript suitable for publication. Please, pay attention to their suggestions, especially those concerning the phylogenetic analysis.**

We look forward to receiving your revised manuscript.

Kind regards,

Giorgio Carnevale, Ph.D

Academic Editor

PLOS ONE

Journal requirements: 1. When submitting your revision, we need you to address these additional requirements. Please ensure that your manuscript meets PLOS ONE's style requirements, including those for file naming. The PLOS ONE style templates can be found at https://journals.plos.org/plosone/s/file?id=wjVg/PLOSOne_formatting_sample_main_body.pdf and https://journals.plos.org/plosone/s/file?id=ba62/PLOSOne_formatting_sample_title_authors_affiliations.pdf. 2. In your manuscript, please provide additional information regarding the specimens used in your study. Ensure that you have reported human remain specimen numbers and complete repository information, including museum name and geographic location.  If permits were required, please ensure that you have provided details for all permits that were obtained, including the full name of the issuing authority, and add the following statement: 'All necessary permits were obtained for the described study, which complied with all relevant regulations.' If no permits were required, please include the following statement: 'No permits were required for the described study, which complied with all relevant regulations.' For more information on PLOS ONE's requirements for paleontology and archeology research, see https://journals.plos.org/plosone/s/submission-guidelines#loc-paleontology-and-archaeology-research. 3. Thank you for stating the following financial disclosure:  [2020, 2021, and 2022 Research Grants of the Association of Earth Science Clubs of Greater Kansas City; 2021 BI Panorama Grant of the department of Ecology & Evolutionary Biology, University of Kansas; 2022 and 2023 EEB Summer Funding, University of Kansas. ].  Please state what role the funders took in the study.  If the funders had no role, please state: ""The funders had no role in study design, data collection and analysis, decision to publish, or preparation of the manuscript."" If this statement is not correct you must amend it as needed. Please include this amended Role of Funder statement in your cover letter; we will change the online submission form on your behalf. 4. Thank you for stating the following in the Acknowledgments Section of your manuscript: [Greatly thanks to H.-P. Schultze ang G. Arratia for the comments and suggestions on this manuscript, and J. Chorn for the suggestions on photographing. I also appreciate A. Stroup and W. Simpson for accessibility to the Field Museum specimens. This work is supported by 2020, 2021, and 2022 Research Grants of the Association of Earth Science Clubs of Greater Kansas City; 2021 BI Panorama Grant of the department of Ecology & Evolutionary Biology, University of Kansas; 2022 and 2023 EEB Summer Funding, University of Kansas. Their support is greatly appreciated.]We note that you have provided funding information that is not currently declared in your Funding Statement. However, funding information should not appear in the Acknowledgments section or other areas of your manuscript. We will only publish funding information present in the Funding Statement section of the online submission form. Please remove any funding-related text from the manuscript and let us know how you would like to update your Funding Statement. Currently, your Funding Statement reads as follows:   [2020, 2021, and 2022 Research Grants of the Association of Earth Science Clubs of Greater Kansas City; 2021 BI Panorama Grant of the department of Ecology & Evolutionary Biology, University of Kansas; 2022 and 2023 EEB Summer Funding, University of Kansas. ].   Please include your amended statements within your cover letter; we will change the online submission form on your behalf. 5. We note that Figure 1 in your submission contain [map/satellite] images which may be copyrighted. All PLOS content is published under the Creative Commons Attribution License (CC BY 4.0), which means that the manuscript, images, and Supporting Information files will be freely available online, and any third party is permitted to access, download, copy, distribute, and use these materials in any way, even commercially, with proper attribution. For these reasons, we cannot publish previously copyrighted maps or satellite images created using proprietary data, such as Google software (Google Maps, Street View, and Earth). For more information, see our copyright guidelines: http://journals.plos.org/plosone/s/licenses-and-copyright. We require you to either (1) present written permission from the copyright holder to publish these figures specifically under the CC BY 4.0 license, or (2) remove the figures from your submission: a. You may seek permission from the original copyright holder of Figure 1 to publish the content specifically under the CC BY 4.0 license.   We recommend that you contact the original copyright holder with the Content Permission Form (http://journals.plos.org/plosone/s/file?id=7c09/content-permission-form.pdf) and the following text:“I request permission for the open-access journal PLOS ONE to publish XXX under the Creative Commons Attribution License (CCAL) CC BY 4.0 (http://creativecommons.org/licenses/by/4.0/). Please be aware that this license allows unrestricted use and distribution, even commercially, by third parties. Please reply and provide explicit written permission to publish XXX under a CC BY license and complete the attached form.” Please upload the completed Content Permission Form or other proof of granted permissions as an ""Other"" file with your submission. In the figure caption of the copyrighted figure, please include the following text: “Reprinted from [ref] under a CC BY license, with permission from [name of publisher], original copyright [original copyright year].” b. If you are unable to obtain permission from the original copyright holder to publish these figures under the CC BY 4.0 license or if the copyright holder’s requirements are incompatible with the CC BY 4.0 license, please either i) remove the figure or ii) supply a replacement figure that complies with the CC BY 4.0 license. Please check copyright information on all replacement figures and update the figure caption with source information. If applicable, please specify in the figure caption text when a figure is similar but not identical to the original image and is therefore for illustrative purposes only.The following resources for replacing copyrighted map figures may be helpful: USGS National Map Viewer (public domain): http://viewer.nationalmap.gov/viewer/The Gateway to Astronaut Photography of Earth (public domain): http://eol.jsc.nasa.gov/sseop/clickmap/Maps at the CIA (public domain): https://www.cia.gov/library/publications/the-world-factbook/index.html and https://www.cia.gov/library/publications/cia-maps-publications/index.htmlNASA Earth Observatory (public domain): http://earthobservatory.nasa.gov/Landsat: http://landsat.visibleearth.nasa.gov/USGS EROS (Earth Resources Observatory and Science (EROS) Center) (public domain): http://eros.usgs.gov/#Natural Earth (public domain): http://www.naturalearthdata.com/ 6. We note that Figure(s) 5, 6, 7, 9, 11, 12, 14 and 15 in your submission contain copyrighted images. All PLOS content is published under the Creative Commons Attribution License (CC BY 4.0), which means that the manuscript, images, and Supporting Information files will be freely available online, and any third party is permitted to access, download, copy, distribute, and use these materials in any way, even commercially, with proper attribution. For more information, see our copyright guidelines: http://journals.plos.org/plosone/s/licenses-and-copyright. We require you to either (1) present written permission from the copyright holder to publish these figures specifically under the CC BY 4.0 license, or (2) remove the figures from your submission: a. You may seek permission from the original copyright holder of Figure(s) 5, 6, 7, 9, 11, 12, 14 and 15 to publish the content specifically under the CC BY 4.0 license.  We recommend that you contact the original copyright holder with the Content Permission Form (http://journals.plos.org/plosone/s/file?id=7c09/content-permission-form.pdf) and the following text:“I request permission for the open-access journal PLOS ONE to publish XXX under the Creative Commons Attribution License (CCAL) CC BY 4.0 (http://creativecommons.org/licenses/by/4.0/). Please be aware that this license allows unrestricted use and distribution, even commercially, by third parties. Please reply and provide explicit written permission to publish XXX under a CC BY license and complete the attached form.” Please upload the completed Content Permission Form or other proof of granted permissions as an ""Other"" file with your submission.  In the figure caption of the copyrighted figure, please include the following text: “Reprinted from [ref] under a CC BY license, with permission from [name of publisher], original copyright [original copyright year].” b. If you are unable to obtain permission from the original copyright holder to publish these figures under the CC BY 4.0 license or if the copyright holder’s requirements are incompatible with the CC BY 4.0 license, please either i) remove the figure or ii) supply a replacement figure that complies with the CC BY 4.0 license. Please check copyright information on all replacement figures and update the figure caption with source information. If applicable, please specify in the figure caption text when a figure is similar but not identical to the original image and is therefore for illustrative purposes only. 7. We are unable to open your Supporting Information file [Suppl Data.nex]. Please kindly revise as necessary and re-upload. 8. Please take this opportunity to be sure you have met all of our guidelines for new species. For proper registration of a new zoological taxon, we require two specific statements to be included in your manuscript.1. In the Results section, the globally unique identifier (GUID), currently in the form of a Life Science Identifier (LSID), should be listed under the new species name, for example: Anochetus boltoni Fisher sp. nov. urn:lsid:zoobank.org:act:B6C072CF-1CA6-40C7-8396-534E91EF7FBBAnother LSID for the manuscript itself should also appear within the Nomenclature statement. You will need to contact Zoobank (zoobank.org/About) to obtain a GUID (LSID). You should receive one LSID for your manuscript and a separate, unique LSID for the new species. 2. Please also insert the following text into the Methods section, in a sub-section to be called ""Nomenclatural Acts"": The electronic edition of this article conforms to the requirements of the amended International Code of Zoological Nomenclature, and hence the new names contained herein are available under that Code from the electronic edition of this article. This published work and the nomenclatural acts it contains have been registered in ZooBank, the online registration system for the ICZN. The ZooBank LSIDs (Life Science Identifiers) can be resolved and the associated information viewed through any standard web browser by appending the LSID to the prefix ""http://zoobank.org/"". The LSID for this publication is: urn:lsid:zoobank.org:pub: XXXXXXX. The electronic edition of this work was published in a journal with an ISSN, and has been archived and is available from the following digital repositories: PubMed Central, LOCKSS [author to insert any additional repositories]. All PLOS ONE articles are deposited in PubMed Central and LOCKSS. If your institute, or those of your co-authors, has its own repository, we recommend that you also deposit the published online article there and include the name in your article.Following a recent ruling by the International Commission on Zoological Nomenclature, electronic journals are now a valid format for publication of new zoological taxa. In order to ensure the valid publication of your new species, please be sure to include the updated version of Nomenclatural Acts (above). A complete explanation of our guidelines for publishing new species can be found on our website: http://www.plosone.org/static/guidelines#zoological.

Reviewers' comments:

Reviewer's Responses to Questions

**Comments to the Author**

1. Is the manuscript technically sound, and do the data support the conclusions?

Reviewer #1: Yes

Reviewer #2: Partly

2. Has the statistical analysis been performed appropriately and rigorously? 

Reviewer #1: Yes

Reviewer #2: Yes

3. Have the authors made all data underlying the findings in their manuscript fully available?

Reviewer #1: Yes

Reviewer #2: Yes

4. Is the manuscript presented in an intelligible fashion and written in standard English?

Reviewer #1: Yes

Reviewer #2: No

5. Review Comments to the Author

Reviewer #1: The manuscript, A new early Actinopterygii from the Mid-Pennsylvanian Logan Quarry Shale member of Indiana, is an important paper to publish because it describes a new taxon from Carboniferous, but more importantly, a new taxon from a locality that has been understudied in the past. It is an important paper to publish but a few changes need to be made and others considered. These changes are detailed below. Please see the attached letter with comments to the author on how to improve the manuscript and a PDF of the manuscript with comment bubbles detailing suggested revisions.

Reviewer #2: This is an interesting description of a new species of Kalops, with lovely camera lucida illustrations. However, despite the detailed descriptive section, the paper needs to be heavily edited before it can be a useful contribution. As currently written, the text is difficult to understand, and there are numerous typos that further obscure clarity and make me a little concerned about the time and attention to detail invested in the manuscript.

The section reporting results of the phylogenetic analysis also contains concerning statements regarding homoplasy. The author does not seem to be clear on the meaning of a synapomorphy vs. a homoplastic trait.

The section on potential habitat and on stratigraphy is underdeveloped. Can the author discuss perhaps the speciation of Kalops given that we now have this species in addition to the Bear Gulch material, in context of the depositional environment and the ages of both deposits? I think the author means to say it’s an epipelagic fish, though the language is unclear, and if it is epipelagic, that says something about its distribution as well as its potential survival through cyclothems.

In general, I think the first few Figures are a little bit under-designed – they would benefit from better mapping of the extent of IB & NAMS. The camera lucida work is beautiful but needs to be labeled in more detail for easy digestion by a reader. Lastly, the phylogeny figure’s OK, but the authors should explain where support values came from, and which support values are hidden in the figure caption.

I have marked numerous comments in the attached PDF using ‘sticky notes,’ so please let me know if those aren’t showing up! Due to time constraints, I didn’t check through the references or re-run the phylogenetic analysis, but the Nexus file of the character matrix looked OK to me when I skimmed it over. I didn’t have time to confirm character codings or check deeply for miscodings or similar issues.

6. PLOS authors have the option to publish the peer review history of their article (what does this mean? ). If published, this will include your full peer review and any attached files.

**Do you want your identity to be public for this peer review?** For information about this choice, including consent withdrawal, please see our Privacy Policy .

Reviewer #1: No

Reviewer #2: No

---

## [Author Response · Author response to Decision Letter 1]

10 Feb 2025

All the answers are included in the "Response to reviewers" file.

---

## [Editor Report · Decision Letter 1]

27 Feb 2025

A new early actinopterygian from the Mid-Pennsylvanian Logan Quarry Shale member of Indiana

PONE-D-24-33140R1

Dear Dr. Shen,

We’re pleased to inform you that your manuscript has been judged scientifically suitable for publication and will be formally accepted for publication once it meets all outstanding technical requirements.

Kind regards,

Giorgio Carnevale, Ph.D

Academic Editor

PLOS ONE

---

## [Editor Report · Acceptance letter]

PONE-D-24-33140R1

PLOS ONE

Dear Dr. Shen,

I'm pleased to inform you that your manuscript has been deemed suitable for publication in PLOS ONE. Congratulations! Your manuscript is now being handed over to our production team.

Kind regards,

on behalf of

Dr. Giorgio Carnevale

Academic Editor

PLOS ONE